# Locking the lattice oxygen in RuO$_2$ to stabilize highly active Ru sites in acidic water oxidation

Xinyu Ping[1,3], Yongduo Liu[1,3], Lixia Zheng[1], Yang Song[1], Lin Guo [2], Siguo Chen [1] ✉ & Zidong Wei [1]

Ruthenium dioxide is presently the most active catalyst for the oxygen evolution reaction (OER) in acidic media but suffers from severe Ru dissolution resulting from the high covalency of Ru-O bonds triggering lattice oxygen oxidation. Here, we report an interstitial silicon-doping strategy to stabilize the highly active Ru sites of RuO$_2$ while suppressing lattice oxygen oxidation. The representative Si-RuO$_2$−0.1 catalyst exhibits high activity and stability in acid with a negligible degradation rate of ~52 μV h$^{-1}$ in an 800 h test and an overpotential of 226 mV at 10 mA cm$^{-2}$. Differential electrochemical mass spectrometry (DEMS) results demonstrate that the lattice oxygen oxidation pathway of the Si-RuO$_2$−0.1 was suppressed by ~95% compared to that of commercial RuO$_2$, which is highly responsible for the extraordinary stability. This work supplied a unique mentality to guide future developments on Ru-based oxide catalysts' stability in an acidic environment.

The proton-exchange membrane water electrolyzer (PEMWE) is regarded as one of the most efficient devices for green hydrogen production because of its small footprint, high current density, and fast response[1–4]. However, the instability of anode catalysts has extremely hindered the large-scale application of PEMWE devices[5–7]. Despite the IrO$_2$ catalyst with the passable stability applied in commercial PEMWE, its unsatisfactory activity asks for a high loading in the anode, sharply raising the cost of PEMWE[8–12]. Fortunately, ruthenium dioxide (RuO$_2$), with low price and high intrinsic activity, is expected to be the ideal alternative to IrO$_2$ after extricating from poor stability[13–15]. Encouragingly, the previously reported works have revealed its degradation mechanism. The high covalency of Ru-O bonds triggers the lattice oxygen oxidation mechanism (LOM), leading to oxygen vacancy (O$_V$) formation and the leaching of active Ru species (soluble RuO$_4$ species), which ultimately accelerates collapse of the crystal structure (Fig. 1)[16–22].

Inspired by the theoretical results, doping heterogeneous metal atoms in RuO$_2$ has been presented and has effectively modified the Ru-O bond covalency[18,23]. Nevertheless, most doping elements, such as Co[24], Na[25], Mn[15], Cr[26], Cu[27] and Ni[28], enhance Ru-O bond covalency while

activating the LOM pathway, resulting in improved activity but poor stability. Very few elements have been reported and used to weaken Ru-O bond covalency and improve the stability of Ru-based oxides. Zhang's group[18] reported that W and Er co-doping increased the formation energy of O$_V$ in RuO$_2$ and prohibited lattice oxygen oxidation by downshifting the O 2$p$ band center away from the Fermi level, eventually enhancing the stability of the doped-RuO$_2$. Ge and coworkers[29] investigated the effect of the doping element electronegativity on Ru-based oxides' electron structure and performance. The prepared SnRuO$_x$ with appropriate Ru-O bond covalency followed the adsorption evolution mechanism (AEM) in the OER and exhibited orders of magnitude lifespan extension compared to that of RuO$_2$. Although the above strategies improved the RuO$_2$ stability somewhat, the doped metal elements are easily subjected to accelerated corrosion in harsh acidic and oxidative environments due to their thermodynamic instability[30,31]. Furthermore, the etched oxide without the doping metals left amorphous RuO$_2$ on the catalyst surface and inevitably started the LOM pathway[30–32]. More importantly, the above-mentioned doped Ru-based oxides are achieved by replacing the Ru

[1]College of Chemistry and Chemical Engineering, State Key Laboratory of Advanced Chemical Power Sources (SKL-ACPS), Chongqing University, Chongqing, China. [2]State Key Laboratory of Catalytic Materials and Reaction Engineering, SINOPEC Research Institute of Petroleum Processing Co., Ltd., Beijing, China. [3]These authors contributed equally: Xinyu Ping, Yongduo Liu. ✉e-mail: csg810519@126.com

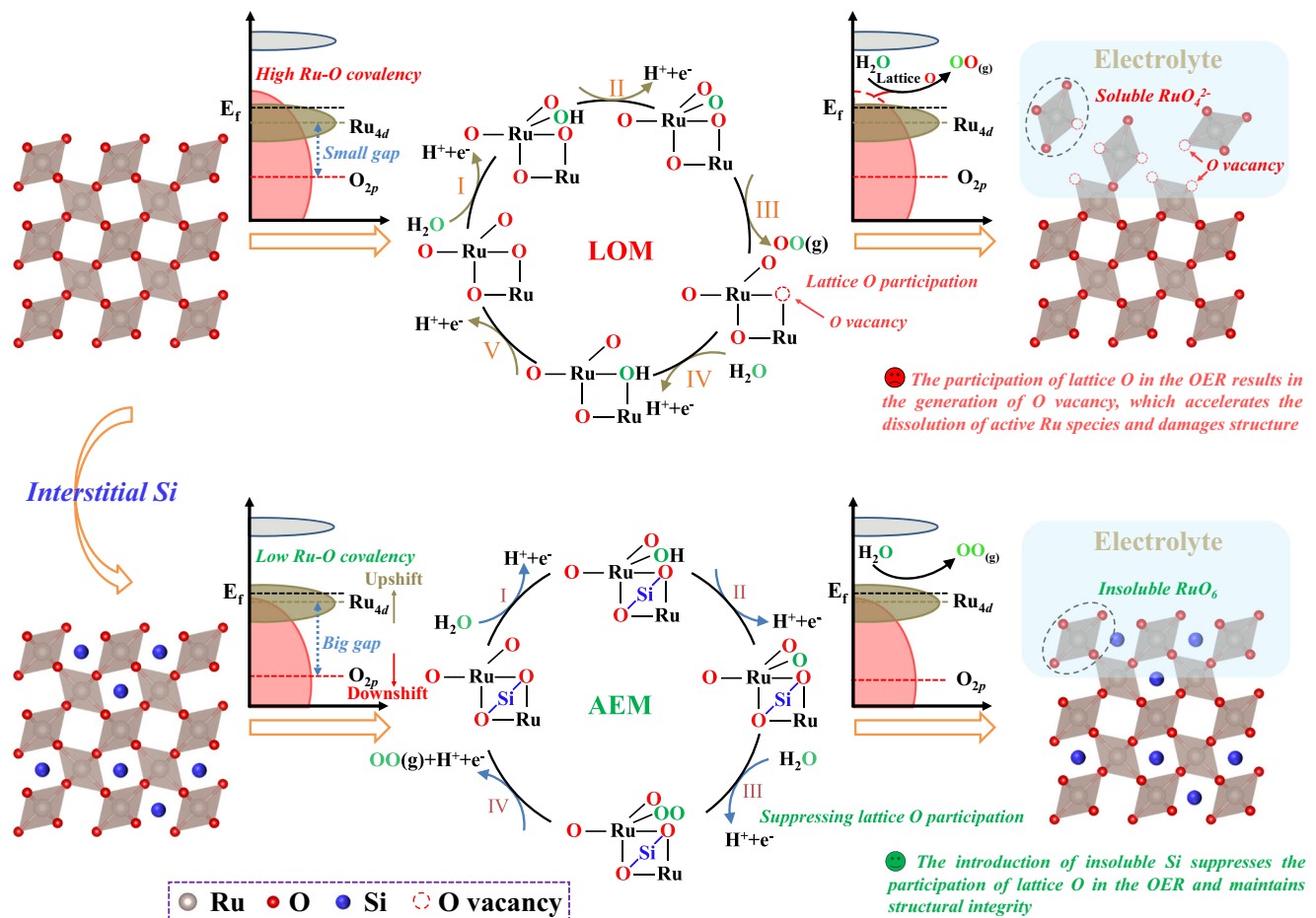

**Fig. 1 | Schematic diagram of the enhanced stability caused by interstitial Si doping.** Crystal structure model of RuO$_2$ (top-left corner) and Si-RuO$_2$ (bottom-left corner). The illustration of LOM pathway of RuO$_2$ (middle top) and AEM pathway of Si-RuO$_2$ (middle top) due to differences in their covalency, in which O labeled in red

represents lattice oxygen and O labeled in green represents oxygen from water. Schematic of the structural changes in RuO$_2$ (top-right corner) and Si-RuO$_2$ (bottom-right corner) under acidic OER conditions.

atoms with doping atoms, which, however, limits the quantity of Ru-O structures in the catalyst and is not beneficial for further improving the activity of Ru-based oxides. Therefore, exploring effective doping strategies for weakening the Ru-O bond covalency without losing the quantity of Ru-O structure is extraordinarily desirable but challenging.

Here, we reveal that metalloid silicon, with superior acid resistance and short effective ion radius (EIR, 0.26 Å for Si$^{4+}$ compared to 0.62 Å for Ru$^{4+}$)[33], is able to be inserted into the interstitial-site of the RuO$_2$ lattice to construct stable RuO$_2$ for the acidic OER. The stability of interstitial Si-doped RuO$_2$ originates from the higher bond dissociation energy of the Si-O bond (798 kJ mol$^{-1}$) compared to that of the Ru-O bond (481 kJ mol$^{-1}$)[33], which decreases the Ru-O bond covalency and is beneficial for suppressing the LOM pathway and strengthening the AEM pathway (Fig. 1). Moreover, the robust Si-O bond can prevent lattice oxygen from participating in the OER and thus prohibit O$_V$ formation. These two factors, in combination with the stability of Si in acidic media, are responsible for the high catalytic stability of the interstitial Si-doped RuO$_2$ catalyst. As a result, a representative Si-RuO$_2$−0.1 catalyst was observed to be stable for at least 800 h with a negligible degradation rate of -52 μV h$^{-1}$ at 10 mA cm$^{-2}$ in acidic electrolyte, outperforming most of the reported Ru-based oxide catalysts (Supplementary Table 4).

## Results and discussions
### Synthesis and characterization of Si-doped RuO$_2$ catalysts
A simple and fast cation-exchange resin (CER) pyrolysis approach was applied to prepare Si-doped RuO$_2$ catalysts with various doping levels

(denoted Si-RuO$_2$-$x$, where $x$ represents the molar ratio of Si to Ru; $x$ = 0, 0.05, 0.1, 0.2, 0.3). The X-ray diffraction (XRD) patterns of all the prepared samples matched well with that of rutile-phase RuO$_2$ (ICSD: 43-1027), demonstrating that the incorporation of Si into the RuO$_2$ lattice did not change the tetragonal phase structure of RuO$_2$ (Fig. 2a). A magnification of the (110) diffraction plane (Fig. 2b), showed that the corresponding diffraction peak gradually shifted toward a lower angle as the Si content increased from 0 to 0.1 and then remained almost unchanged as the Si content further increased from 0.1−0.3. Considering that both the EIR of Si$^{4+}$ and its coordination number (CN) with oxygen in nature (EIR = 0.26 Å, CN = 4) were much lower than those of Ru$^{4+}$ (EIR = 0.62 Å, CN = 6), we inferred that Si tends to interstitially insert into the RuO$_2$ lattice rather than replace the Ru atoms[33–37]. To visually prove that Si was inserted into the RuO$_2$ interstice, spherical aberration−corrected HAADF-STEM measurements were performed. As shown in Fig. 2c, the lattice fringes with interplanar spacings of 0.318 nm and 0.254 nm were assigned to the (110) and (101) planes of rutile RuO$_2$, respectively. Furthermore, some isolated Si atoms with low imaging contrast, which is characteristic of light elements with lower atomic numbers, were also observed in the lattice interstices of RuO$_2$ (Fig. 2d-e). This assertion was confirmed by atomic line profiles analysis (Fig. 2f-i). Based on the above analysis, we undoubtedly concluded that Si inserts into the RuO$_2$ interstice.

The morphology and elemental distribution of different Si-RuO$_2$-$x$ catalysts were further investigated by transmission electron microscopy (TEM) and energy-dispersive X-ray spectroscopy (EDS)

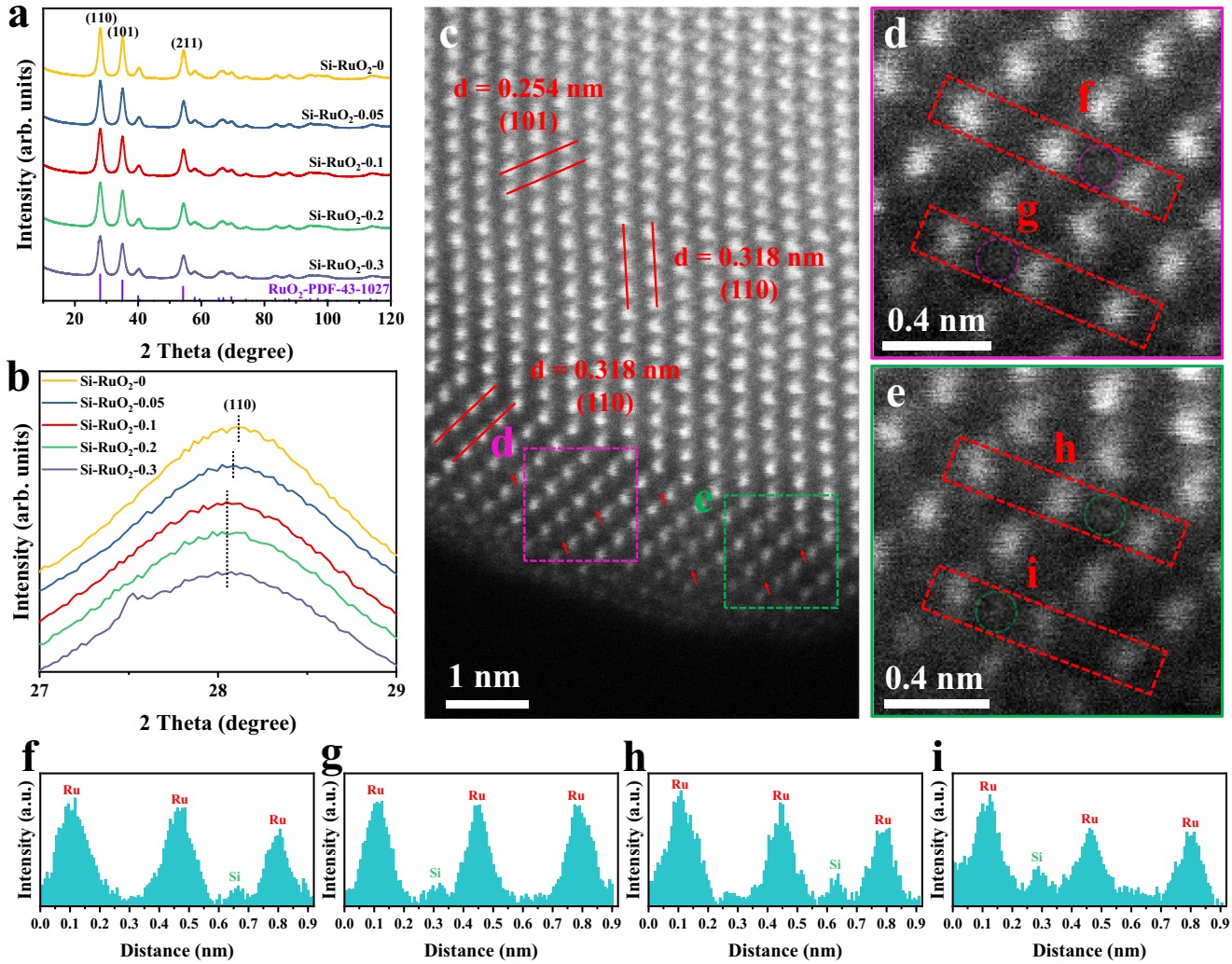

**Fig. 2 | Phase and morphology characterizations. a** XRD patterns of all Si-RuO$_2$-$x$ samples. **b** Magnified image of the (110) diffraction peaks of Si-RuO$_2$-$x$. **c** HAADF-STEM image of Si-RuO$_2$−0.1. **d, e** Magnified HAADF-STEM image obtained from the area highlighted with purple and green in Fig. 1c. **f–i** Line-scanning intensity profile obtained from the area highlighted with red lines in (**e, f**).

elemental mapping. As displayed in Supplementary Figs. 1a, b–5a, b, the TEM images of all Si-RuO$_2$-$x$ catalysts presented a typical porous structure composed of nanoparticles with an average diameter of 5-6 nm. The formation of such a structure could be explained by the CER acting as a skeleton structure to prevent catalyst agglomeration and as a soft template for pore formation during the annealing process[38]. The high-resolution TEM (HR-TEM) images showed that the lattice fringe spacings of the (110) and (101) planes were not significantly different among these catalysts (Supplementary Figs. 1c–5c), further confirming that after Si intercalation, the original morphology and rutile structure of RuO$_2$ were still preserved. The corresponding EDS mapping showed that Si, Ru and O were uniformly distributed throughout the entire catalyst at the low Si doping levels of Si-RuO$_2$−0.05, Si-RuO$_2$−0.1 and Si-RuO$_2$−0.2 (Supplementary Figs. 2d–4d). However, when the Si content increased to 0.3, some Si and O atoms were highly concentrated in the interior of the sample, indicating that some Si atoms did not enter the RuO$_2$ interstice and instead formed insulating SiO$_2$ (Supplementary Fig. 5d). Combined with the XRD results, it was inferred that the ideal doping level of Si in RuO$_2$ interstices was around 10%.

The influence of doped Si on the electronic properties of Ru and O was investigated by X-ray photoelectron spectroscopy (XPS). As presented in the Si 2$p$ spectra in Fig. 3a, the Si-RuO$_2$−0.05 and Si-RuO$_2$−0.1 samples presented only one prominent peak at 102.4 eV,

which was attributed to the Si-O bond associated with interstitial Si. When the Si content increased to 0.2 and 0.3, an additional peak at 104.1 eV consistent with the Si-O bond in the SiO$_2$ phase appeared. This result confirmed that the actual Si solubility in the RuO$_2$ lattice was limited to around 10% and that the extra amount of Si preferentially formed an amorphous SiO$_2$ phase, which agrees with the XRD and EDS results. In addition to considering the Si 2$p$ spectra, we also analyzed the O 1$s$ and Ru 3$p$ spectra (Fig. 3b, c). The peak of lattice oxygen positively shifted toward a higher binding energy with increasing Si content, while that of Ru shifted toward a lower binding energy ($x \le 0.1$) and then remained almost unchanged ($0.1 \le x \le 0.3$). The positive shift of the O 1$s$ binding energy and negative shift of the Ru 3$p$ binding energy provided direct evidence for the weakening of the Ru-O bond covalency by the incorporation of Si into the RuO$_2$ interstices[39–41].

The valence state and coordination environment of Ru in the representative Si-RuO$_2$−0.1 sample were further revealed by X-ray absorption fine structure (XAFS) analysis. The Ru $K$-edge X-ray absorption near-edge structure (XANES) spectra showed that the absorption edge of Si-RuO$_2$−0.1 was slightly negatively shifted compared to that of commercial RuO$_2$ (Com-RuO$_2$) (Fig. 3d and Supplementary Fig. 6), indicating that Si-RuO$_2$−0.1 had a lower Ru valence state than Com-RuO$_2$. The corresponding Fourier-transformed

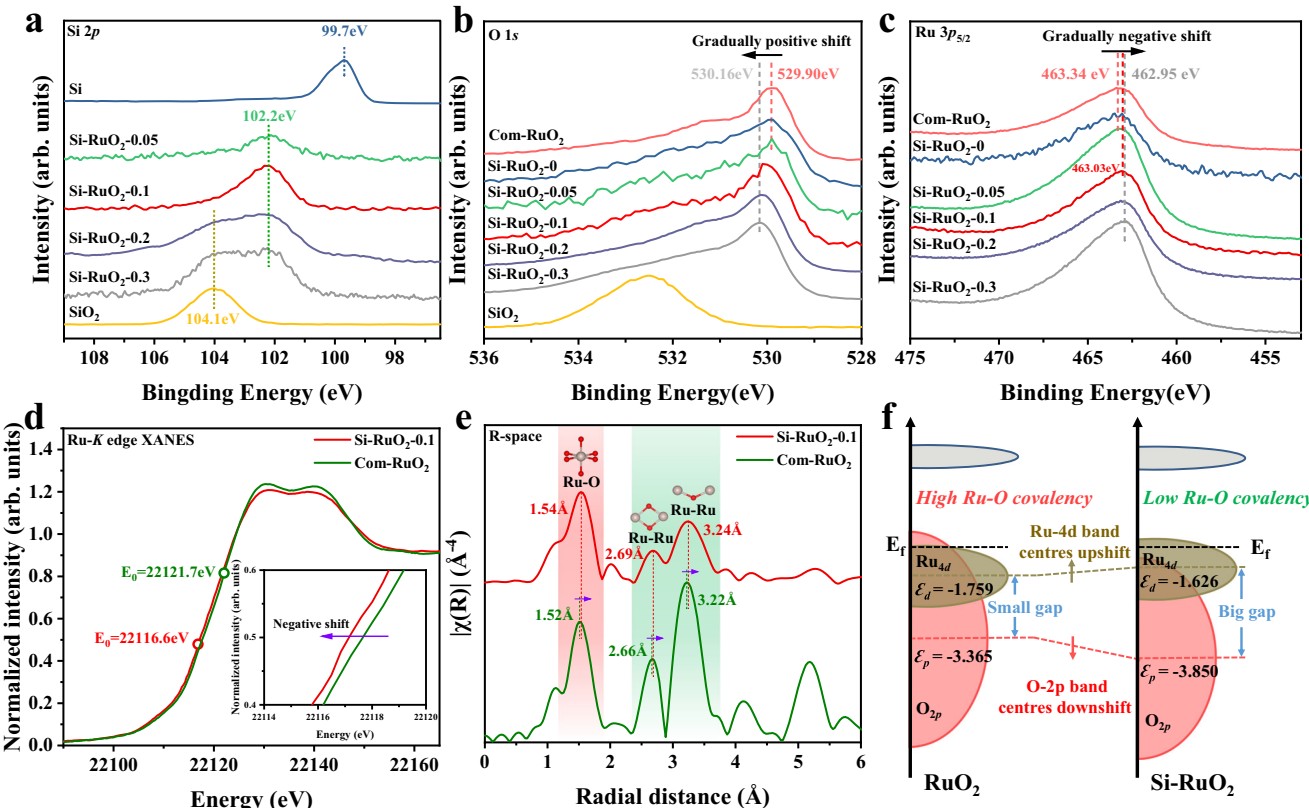

**Fig. 3 | Electronic structure characterizations. a–c** XPS spectra of Si 2p, O 1s and Ru 3p$_{5/2}$, respectively. **d** Normalized Ru K-edge XANES spectra of Si-RuO$_2$–0.1 and Com-RuO$_2$. **e** Fourier-transform EXAFS spectra of Si-RuO$_2$–0.1 and Com-RuO$_2$.

**f** Schematic diagram of the band structures of Si-RuO$_2$ and RuO$_2$ based on Ru and O atoms on the (110) plane.

extended X-ray absorption fine structure (FT-EXAFS) spectra of Si-RuO$_2$–0.1 and Com-RuO$_2$ showed that the predominant peak of the Ru-O scattering path appeared at 1.54 Å for Si-RuO$_2$–0.1, and this peak was slightly contracted to 1.52 Å in Com-RuO$_2$ (Fig. 3e), suggesting that the Si tuned to Ru-O bond length. This phenomenon was also observed when comparing the Ru-Ru scattering paths in Si-RuO$_2$–0.1 and Com-RuO$_2$ (Fig. 3e). To more specific, the FT-EXAFS spectra were reasonably fitted (Supplementary Fig. 7). The best-fit results confirmed that Si doping led to the elongation of the Ru-O bond from 1.96 Å to 1.98 Å (Supplementary Table 1). The lower Ru valence state and elongated Ru-O bonds further proved that the Ru-O bond covalency was weakened by introducing Si into RuO$_2$ interstices[42–44]. To further verify this conclusion, the partial density of states (PDOS) calculations on the (110) facet and bulk of RuO$_2$ and Si-RuO$_2$ model (see Methods, Supplementary Fig. 8-9 and Supplementary Table 2) were performed, respectively. As interpreted in Fig. 3f and Supplementary Fig. 10a, after introducing Si into the interstitial sites, the surface Ru 4d band center ($\varepsilon_d$) upshifted from −1.759 eV − −1.626 eV, while the surface O 2p band center ($\varepsilon_p$) downshifted from −3.365 eV − −3.850 eV, suggesting that the gap between $\varepsilon_d$ and $\varepsilon_p$ was obviously enlarged. Besides, the corresponding calculated this gap for RuO$_2$ bulk and Si-RuO$_2$ bulk were 1.409 eV and 1.844 eV (Supplementary Fig. 10b). All results indicated that the covalency of Ru-O bond in Si-RuO$_2$ was lower than that of RuO$_2$. It is generally agreed that weak metal−oxygen covalency can promote the stability of catalysts by suppressing the LOM pathway during the OER[20,45–47].

## OER performance measurement and the origin of the enhanced activity

To explore the influence of the Si content on the OER activity, we recorded the polarization curves of as-prepared Si-RuO$_2$-x and Com-

RuO$_2$ in an O$_2$-saturated 0.1 M HClO$_4$ solution. As shown in Fig. 4a-b, the Si-RuO$_2$–0 sample without Si exhibited an overpotential of 248 mV at 10 mA cm$^{-2}$, which was superior to that of Com-RuO$_2$ (291 mV). This improved OER activity was attributed to the porous structure and smaller nanoparticles providing more active sites for the OER. With increasing Si content, the overpotential gradually decreased and reached a minimum of 226 mV for Si-RuO$_2$–0.1; however, this minimum value increased to 234 mV for Si-RuO$_2$–0.2 and 238 mV for Si-RuO$_2$–0.3. To gain insight into the causes of this activity trend, we analyzed the Tafel slope and electrochemical impedance spectroscopy (EIS) of these catalysts to evaluate their reaction kinetics and charge-transfer ability. As displayed in Fig. 4c, the Tafel slope of Si-RuO$_2$–0.1 was 33.0 mV dec$^{-1}$, which was smaller than that of the other catalysts, indicating that Si-RuO$_2$–0.1 had the fastest reaction kinetics. The EIS plots in Fig. 4d show that Si-RuO$_2$–0.1 had the smallest semicircle radius among these fabricated samples, indicating that the charge-transfer resistance between the Si-RuO$_2$–0.1 catalyst and reactants was the smallest. These results suggested that incorporating an appropriate Si content (x ≤ 0.1) into the RuO$_2$ matrix effectively enhanced the reaction rate and charge transfer, thus enhancing the intrinsic activity. However, an excessive Si content (x > 0.1) resulted in the formation of an insulating SiO$_2$ phase that blocked the charge transfer across the grains and covered the catalytic active sites of RuO$_2$[34,48,49].

To comprehensively understand the origin of the enhanced acidic OER performance of the Si-RuO$_2$–0.1 catalyst, we carried out DFT calculations to investigate the change in the Gibbs free energy of the reaction intermediates based on the four elementary steps in OER process (Fig. 4e). As presented in Fig. 4f, Supplementary Fig. 11 and Supplementary Table 3, the formation of *OOH intermediates were the rate-determining step (RDS) for RuO$_2$ and Si-RuO$_2$, which is in good agreement with previous reports[19,26,50,51]. When the input potential was

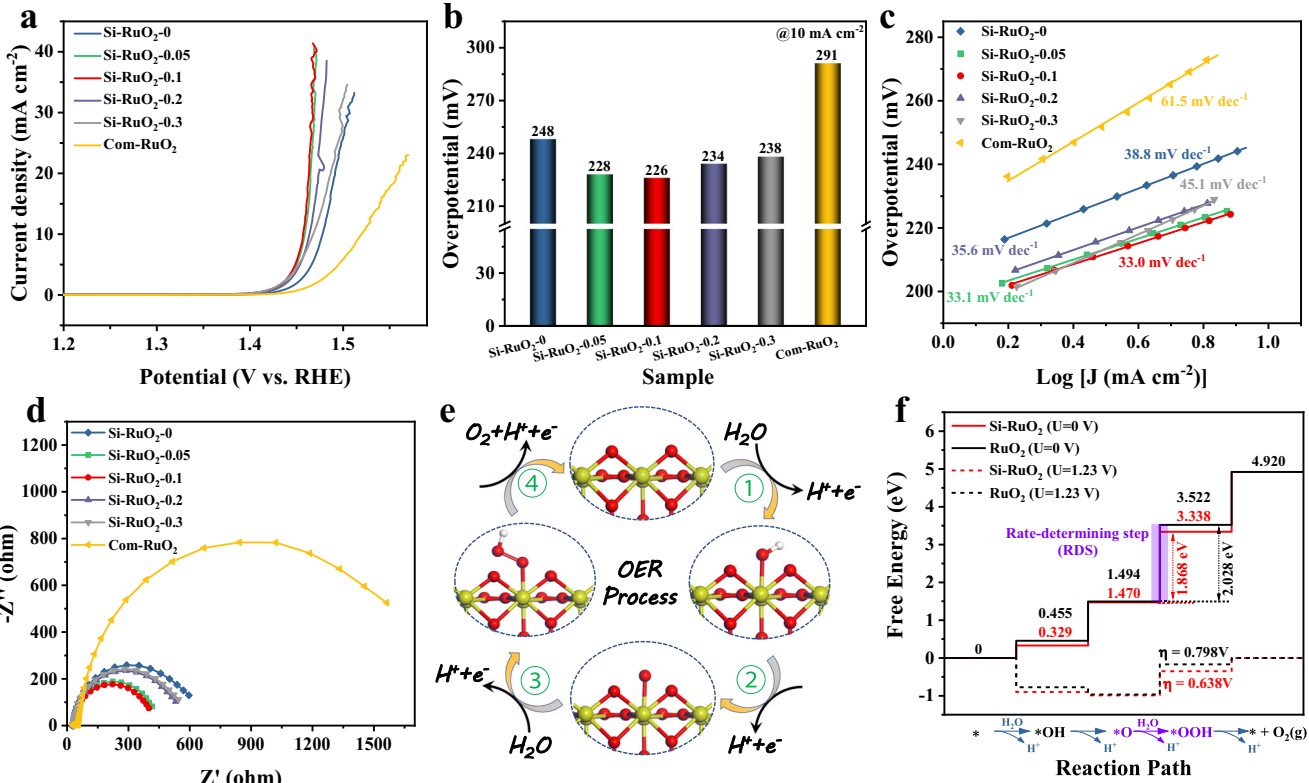

**Fig. 4 | The catalytic activity for OER in 0.1M HClO₄ (pH = 0.108). a** Polarization curves of Si-RuO₂-*x* and Com-RuO₂ (with a loading of 0.30 mg_cat cm⁻²). **b** Overpotential at 10 mA cm⁻²_geo of different catalysts from (**a**). **c** Corresponding Tafel slopes calculated from (**a**). **d** EIS plots of different catalysts. The high- frequency region of Nyquist plot is used to determine the solution resistance (about 30 Ω) for iR-correction. **e** The four electrons mechanism of OER in acidic solution. **f** Calculated energy barrier diagrams of Si-RuO₂ and RuO₂.

0 V, the free energy barrier for RuO₂ is 2.028 eV, and the corresponding limiting overpotential is 0.798 V, which agrees well with previous calculations[14,18,26,27]. However, compared with RuO₂, Si-RuO₂ exhibited a lower free energy barrier (1.868 eV) and overpotential (0.638 V). These results revealed that incorporating Si into RuO₂ interstices plays a critical role for boosting OER activity.

**Catalytic stability evaluation**
Apart from activity, stability is another major concern for evaluating the industrial application of electrocatalysts[52,53]. As presented in Fig. 5a, the synthesized Si-RuO₂-0 and Com-RuO₂ catalysts only ran for 40 h and 18 h at 10 mA cm⁻², respectively, and then rose sharply to 2.5 V. In stark contrast, the Si-RuO₂-0.1 catalyst was active for a record 800 h with only a 42 mV increase in overpotential (Fig. 5a). The calculated degradation rate was -52 μV h⁻¹, outperforming most of the reported Ru-based oxide catalysts in acidic media (Fig. 5b and Supplementary Table 4). An inductively coupled plasma–mass spectrometry (ICP–MS) experiment was performed to monitor the Ru ion dissolution amount in the electrolyte during the OER process. As shown in Fig. 5c, the amount of dissolved Ru ions observed for Si-RuO₂-0 was far greater than that observed for Com-RuO₂, although Si-RuO₂-0 remained active for a longer time. The severe leaching of Ru observed for Si-RuO₂-0 was attributed to its larger surface area, which resulted in more exposed Ru sites on the reaction interface. However, with the introduction of Si, the dissolution amount and rate of Ru ions significantly decreased, suggesting that Si played an important role in inhibiting Ru ion dissolution. The so-called stability number (S-number) was calculated based on the amounts of dissolved Ru ions and generated O₂ to assess the stability of the catalysts[6,54]. Note that for each sampling time point, the S-number of the Si-RuO₂-0.1 catalyst was the largest and increased with the reaction time (Supplementary

Fig. 12). The S-number of the Si-RuO₂-0.1 catalyst after the 13 h OER test was $3.42 \times 10^4$, which was 8.7-fold and 2.5-fold that of Si-RuO₂-0 ($3.95 \times 10^3$) and Com-RuO₂ ($1.39 \times 10^4$), respectively. These results indicated that the incorporation of Si into RuO₂ interstices enhanced the stability of RuO₂ toward the acidic OER.

**Characterization of Si-RuO₂−0.1 after acidic OER**
Furthermore, a series of characterizations, including XRD, TEM and XPS, were performed for the spent Si-RuO₂−0.1 catalysts to investigate the structural evolution of Si-RuO₂-0.1. Obviously, the crystalline structure and morphology of Si-RuO₂−0.1 still maintained its integrity after the 800 h stability test (Fig. 6a-b and Supplementary Fig. 13). Meanwhile, the Si, Ru and O elements were also uniformly distributed in Si-RuO₂-0.1, further illustrating the good stability of Si-RuO₂-0.1 toward acidic OER (Fig. 6c). To prove that the introduction of Si highly improved dissolution and oxidation resistance of RuO₂ toward the acidic OER, the chemical state changes for Ru and O in Si-RuO₂−0.1 before and after the 24 h stability test were further investigated and compared with those of Si-RuO₂−0 and Com-RuO₂ (Caution: The spent Com-RuO₂ sample was only tested for 18 h). For the Ru 3*p* spectra of Si-RuO₂−0.1, the Ru$^{>+4}$/ Ru$^{+4}$ value increased from 0.34 – 0.39 after the stability test, indicating the inevitable oxidation of catalysts under a high anode potential[9,55]. Despite this, for the Si-RuO₂−0 and Com-RuO₂ samples, the change in the Ru$^{>+4}$/ Ru$^{+4}$ value is more significant, increasing from 0.43 – 0.58 and from 0.38 to 0.49, respectively (Fig. 6d and Supplementary Fig. 14). This result revealed that the introduced Si kept the Ru from overoxidation during the OER. Likewise, the evolution of oxygen species is also revealed by combining O 1*s* spectra before and after the stability test. As displayed in Fig. 6e and Supplementary Fig. 15, the remarkable increase in O_V/O_L value for Si-RuO₂−0

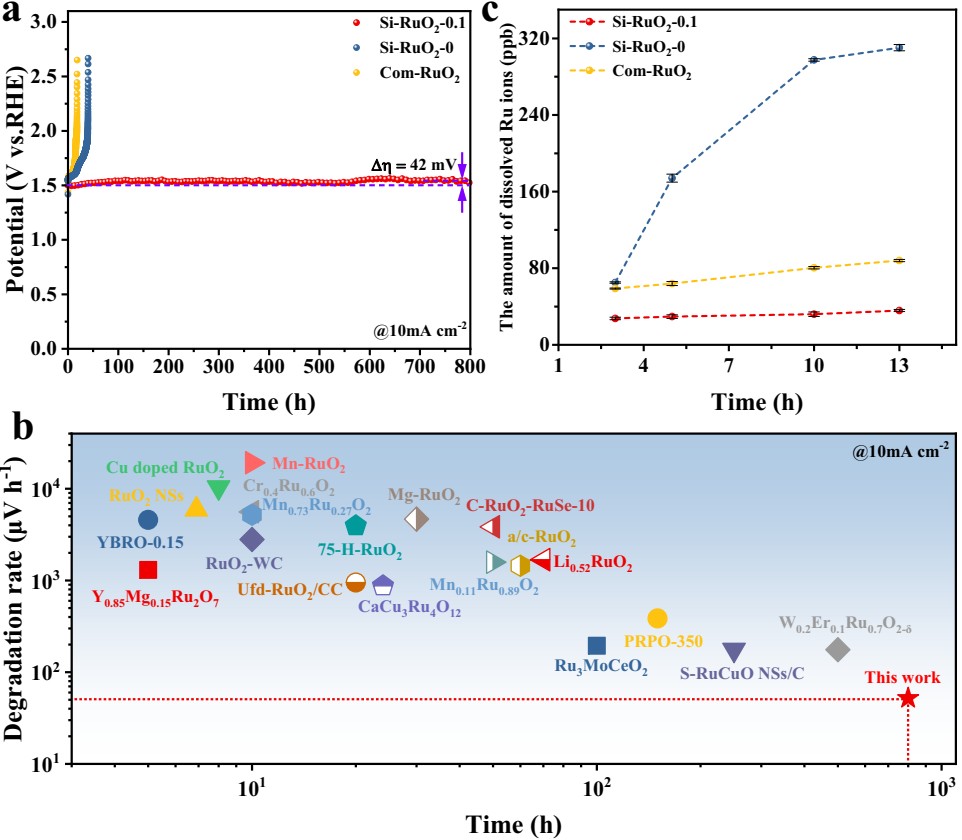

**Fig. 5 | Catalytic stability for OER in 0.1M HClO₄ (pH = 0.108).**
**a** Chronopotentiometry stability curves of Si·RuO₂·0.1 and the control samples at 10 mA cm⁻². The stability curves are collected with a catalyst loading of 1.5 mg_cat cm⁻². The solution resistance is ~4.6 Ω in this stability test system. **b** Comparison of the stability and degradation rate of the Ru-based oxide catalyst at 10 mA cm⁻² with that of recently reported catalysts in acidic solution. **c** Comparison of the amounts of leached Ru ions during the OER process observed for Si·RuO₂−0.1, Si·RuO₂−0 and Com·RuO₂.

and Com·RuO₂ suggests that the lattice O is involved in O₂ generation to a large extent, which will accelerate the dissolution of active Ru species[16,56]. In contrast, the O_V/O_L value was only slightly increased from 1.37 − 1.41 for Si·RuO₂·0.1, indicating that the AEM pathway dominated the OER process, rather than the LOM pathway. This assertion was confirmed by electron paramagnetic resonance (EPR) (Fig. 6f), in which the signal intensity of O_V at around 3513 G ($g$ = 2.001) showed no obvious change[57–59].

**DEMS measurement**

The above analysis indicated that the introduction of Si into RuO₂ was capable of suppressed the LOM pathway during the OER process[60]. To validate this hypothesis from an experimental perspective, we carried out differential electrochemical mass spectrometry (DEMS) using heavy-oxygen water (H₂¹⁸O) to detect the level of participation of lattice oxygen atoms during the OER process. The catalyst was first labeled by cyclic voltammetry (CV) in a 0.1 M HClO₄ solution containing H₂¹⁸O, and then the evolved O₂ was measured in a 0.1 M HClO₄ solution of H₂¹⁶O (see Fig. 7a and Methods). The signals of the evolved ³⁴O₂ reflected the direct ¹⁶O-¹⁸O coupling of ¹⁸O in the lattice and ¹⁶O in water[17]. As presented in the experimental results, the ratio of ³⁴O₂ to (³²O₂ + ³⁴O₂) was only 0.334% for Si·RuO₂−0.1, which was only one-twentieth that of Com·RuO₂ (6.72%) (Fig. 7b–d and Supplementary Fig. 16), indicating that the contribution of lattice oxygen to the OER was suppressed by ~95% on Si·RuO₂−0.1 compared to Com·RuO₂. Conclusively, the LOM pathway of the Si·RuO₂−0.1 sample was greatly hindered, which is also giving strong support for its enhanced acidic OER stability.

In summary, we present a type of interstitial silicon-decorated RuO₂ catalyst that exhibits high activity and stability toward the acidic OER. Si incorporation into the RuO₂ lattice reduced the energy barrier of the RDS by optimizing the adsorption strength of *OOH intermediates onto active Ru sites, thus promoting OER performance. More importantly, we found that the robust Si-O bond formed by inserting acid-resistant Si into RuO₂ interstices was able to weaken Ru-O bond covalency. Under the combined action of acid-resistant Si, strong Si-O interactions and weak Ru-O bond covalency, interstitial silicon-decorated RuO₂ catalysts can suppress the LOM pathway in the OER for a long time, thereby exhibiting high stability. This work sheds light on the design of advanced catalysts with favorable stability toward the OER.

## Methods
### Chemicals and materials
Ruthenium chloride hydrate (RuCl₃·xH₂O, 37.5 wt% Ru) and tetraethyl orthosilicate (TEOS, Si(OC₂H₅)₄) were purchased from Chengdu Chron Chemical Reagent Limited Corporation (Chengdu, China). Commercial ruthenium oxide (RuO₂) (≥ 99.99%) was obtained from Sigma–Aldrich. A D113-type cation exchange resin (CER) was purchased from Tianjin Hongbomei Chemical Technology Co., Ltd. (Tianjin, China). Hydrochloric acid (HCl), sodium hydroxide (NaOH), sodium chloride (NaCl) and ethanol (C₂H₆O) were provided by Sinopharm LTD. The adsorbed water was removed from hydrous RuCl₃ in a vacuum oven at 120 °C for 24 h, and then the RuCl₃ was stored in a glove box with argon gas. The deionized water (DI) used in all experiments was obtained from a Millipore system.

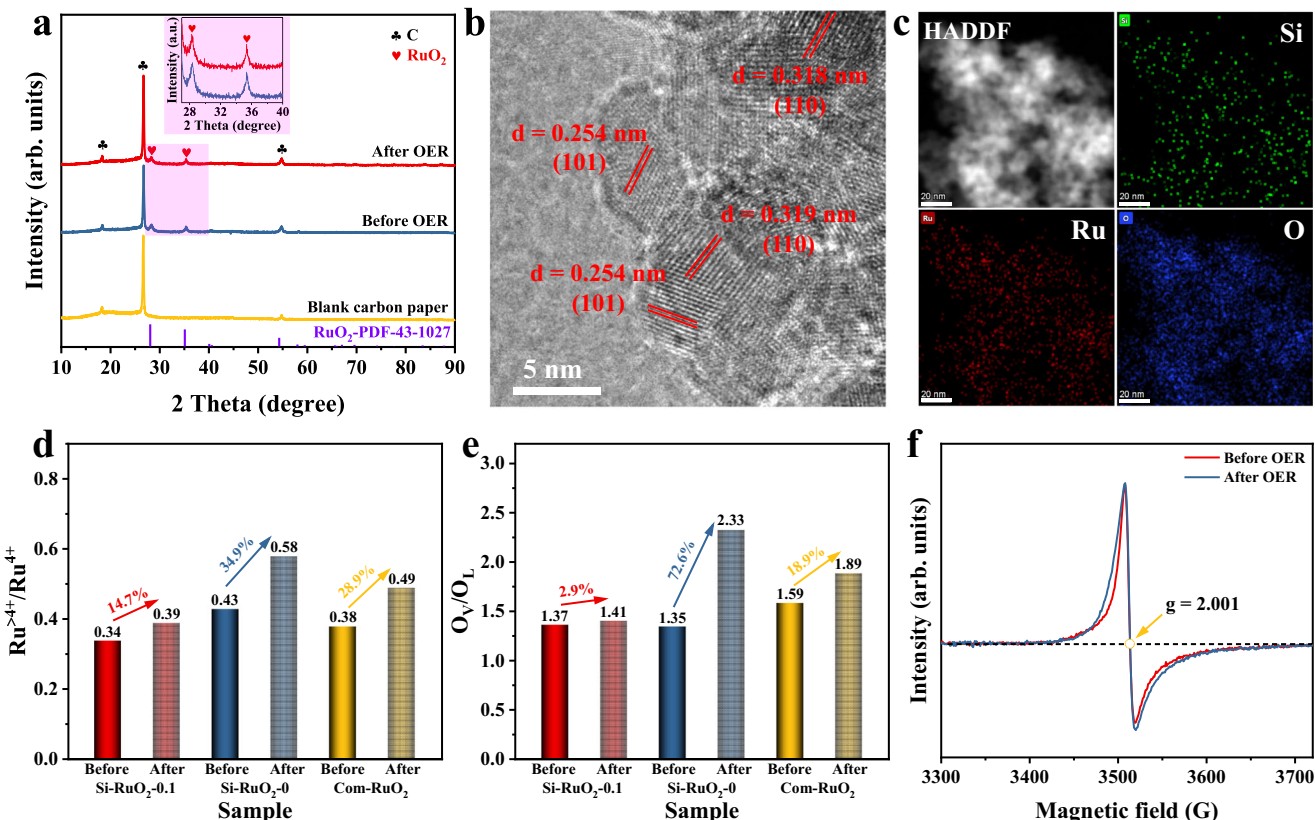

**Fig. 6 | Morphology and structure characterizations for Si·RuO₂−0.1 after stability test. a** XRD patterns. **b** HR-TEM images, and **c** corresponding mapping images for Si·RuO₂−0.1 after the 800 h OER stability test. **d-e** Ru$^{>+4}$/Ru$^{+4}$ and O$_V$/O$_L$ ratios for Si·RuO₂-0.1, Si·RuO₂−0 and Com·RuO₂ before and after 24 h of stability. **f** EPR spectra of Si·RuO₂−0.1 before and after the stability test.

## Preparation of Si·RuO₂-$x$ ($x$ = 0, 0.05, 0.1, 0.2, 0.3) catalysts

Typically, TEOS (334 μL) was dissolved in 30 mL of C₂H₆O to form a 0.05 mmol mL$^{-1}$ TEOS solution. A total of 68 mg (or 0.25 mmol Ru) of dry RuCl₃·$x$H₂O powder was dissolved in 1 mL ultrapure water. Subsequently, 500 μL of 0.05 mmol mL$^{-1}$ TEOS solution and the above RuCl₃ solution were added to 1.5 g of powder-like CER in turn and then thoroughly ground for 30 min. After 2 h of rest, the as-obtained powder with TEOS and RuCl₃ was dried at 60 °C for 8 h and then subjected to calcination in air under ambient pressure at 450 °C for 8 h. After the furnace cooled to room temperature, the black products were collected. The obtained products were washed several times with 40 mL of DI water at 80 °C and then dried in a vacuum oven at 60 °C to finally obtain the Si·RuO₂−0.1 sample. To synthesize the Si·RuO₂−0, Si·RuO₂−0.05, Si·RuO₂−0.2 and Si·RuO₂−0.3 samples, the same procedure with the Si·RuO₂−0.1 sample was used by changing the additive amount of TEOS solution (0 μL, 250 μL, 1000 μL and 1500 μL, respectively).

## Material characterization

Powder X-ray diffractometer (XRD) was performed on a PANalytical X'pert with Cu K$\alpha$ radiation ($\lambda$ = 1.542 Å) at room temperature to obtain the crystalline structure of the samples. The X-ray photoelectron spectroscopy (XPS) signals of the samples were collected with an ESCALAB250Xi spectrometer with an Al K$\alpha$ light source (Al K$\alpha$, 1.4866 keV). For transmission electron microscopy (TEM), a FEI Talos F200S instrument was used to characterize the microstructure of the samples under an accelerating voltage of 120 kV, and corresponding energy dispersive X-ray (EDS) mapping was employed to identify the element composition and distribution. A spherical-aberration-corrected transmission electron microscope (JEM-ARM200F) was used to identify the position of Si in RuO₂. Electron paramagnetic

resonance (EPR) spectra were obtained on a Bruker EMXPLUS spectrometer with a microwave frequency of 9.84 GHz. The $K$-edge X-ray absorption spectra (XAS) of Ru were recorded at the BL14W1 beamline of the Shanghai Synchrotron Radiation Facility (SSRF) (Shanghai, China).

## Electrochemical measurements

A conventional three-electrode system in Gamry electrochemical workstation (Reference 3000) was employed to evaluate the electrochemical performance of the samples. Ag/AgCl (3.5 M KCl-saturated) and graphite rods ($\Phi$ = 6 mm) served as the reference electrode (RE) and counter electrode (CE), respectively. The catalyst ink was prepared by dispersing 4 mg of catalyst into a mixture of 1 mL isopropyl alcohol and 15 μL Nafion solution (5 wt.%), followed by ultrasonic dispersion. Then, 15 μL of the abovementioned ink was dropped onto a cleaned glassy carbon (GC) electrode ($\Phi$ = 5 mm) and dried under an infrared lamp to form the working electrode (WE) with the catalysts. In all experiments, the electrolyte was 0.1 M perchloric acid (HClO₄) solution. Before testing, the Ag/AgCl electrode was calibrated by cyclic voltammetry (CV) using a purified Pt mesh as the WE in H₂-saturated 0.1 M HClO₄ electrolyte, and the average voltage value was recorded as $E_c$ when the current was zero. The value of $E_c$ was generally between 0.260 V and 0.270 V for the Ag/AgCl electrode in 0.1 M HClO₄ solution. All potentials were calibrated relative to the reversible hydrogen electrode (RHE) with iR compensation, according to the following calculations:

$$E_{RHE} = E_{Ag/AgCl} + E_c - I_{mea} \times R_{sol} \qquad (1)$$

where $E_{Ag/AgCl}$ is the potential relative to the Ag/AgCl electrode, which is the set potential during all measurements, and $E_c$ is the potential of

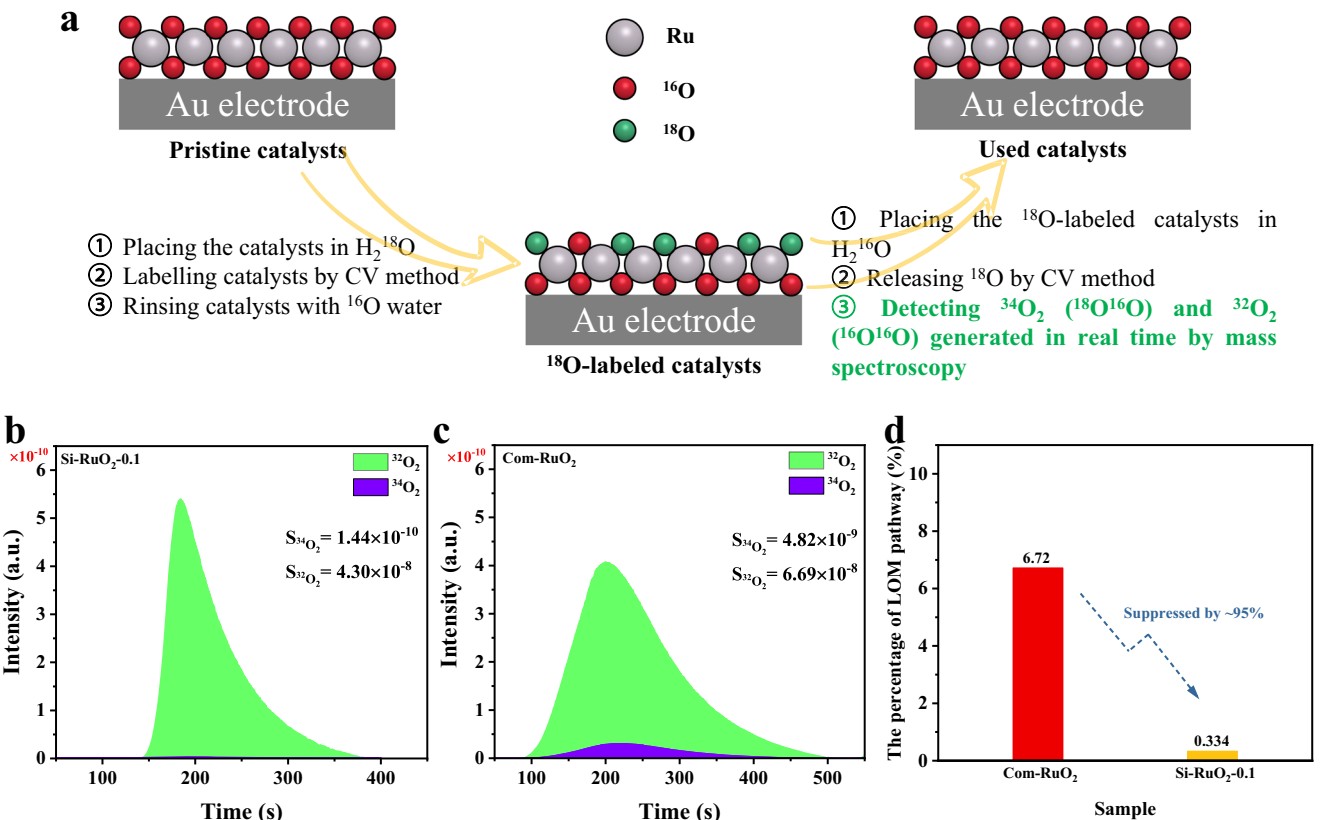

**Fig. 7 | DEMS measurement. a** Schematic diagram of the DEMS measurement process. **b-c** DEMS signals of $^{34}O_2$ ($^{16}O^{18}O$) and $^{32}O_2$ ($^{16}O^{16}O$) from the produced oxygen gas for the $^{18}O$-labeled Si-RuO$_2$−0.1 (**b**) and Com-RuO$_2$ (**c**) catalysts in 0.1 M HClO$_4$ electrolyte containing H$_2^{16}$O. **d** Percent contribution of the LOM pathway in the OER.

the Ag/AgCl electrode relative to the RHE. $I_{mea}$ is the measured polarization current. $R_{sol}$ is the solution resistance; the solution resistance of this test system was ~30 ohms.

To assess the true OER performance, the WEs were first subjected to 50 cycles of CV between 1.0 and 1.6 V (vs. RHE) at a scan rate of 50 mV s$^{-1}$ to stabilize the catalysts in an O$_2$-saturated 0.1 M HClO$_4$ solution. Then, linear sweep voltammetry (LSV) was used to measure the OER polarization curve from 1.0 to 1.7 V (vs. RHE) at a sweep rate of 5 mV s$^{-1}$ with a 1600 rpm rotation speed. Electrochemical impedance spectroscopy (EIS) was obtained in the frequency range from 10$^5$ Hz to 10$^{-2}$ Hz at a bias voltage of 1.4 V (vs. RHE) with a 10 mV amplitude. To investigate the stability of the catalysts, carbon paper (CP, with a surface area of 1 cm$^2$) with 1.5 mg of catalyst was employed as the WEs, and then chronopotentiometry was used to record the E-t curve at a current density of 10 mA cm$^{-2}$.

**Inductively coupled plasma–mass spectrometry (ICP–MS) analysis of Ru ion dissolution**
The ICP–MS experiments were also carried out a NexION 5000 from Perkin Elmer to quantify the dissolution of Ru ions for Si-RuO$_2$−0.1, Si-RuO$_2$−0 and Com-RuO$_2$ during the OER process. Carbon paper with 1.5 mg cm$^{-2}$ catalysts was used as the WE, and electrolysis was performed at 10 mA cm$^{-2}$ in 100 mL of 0.1 M HClO$_4$ solution. Aliquots of 4 mL of electrolyte were removed after 3 h, 5 h, 10 h and 13 h of electrolysis and replaced with 4 mL of fresh electrolyte. The electrolyte aliquots were directly subjected to elemental analysis by ICP–MS.

Based on the ICP–MS results, the stability number (S-number) was also calculated using the following equation:

$$S-number = \frac{N_{O_2}}{N_{dis}} \quad (2)$$

where $N_{O_2}$ is the total amount of evolved oxygen and $N_{dis}$ is the total amount of dissolved Ru ions according to the ICP–MS results.

**Differential electrochemical mass spectrometry (DEMS)**
In situ DEMS involving heavy-oxygen water (H$_2^{18}$O) was performed to identify the level of participation for lattice oxygen during the OER process in a QAS 100 device (see ref. 60. for details). The catalysts were dripped onto a porous gold (Au) disk electrode with a catalyst loading of 0.3 mg cm$^{-2}$. The porous Au disk electrode with catalysts, Ag/AgCl electrode and pure Pt wire were used as the WE, RE and CE, respectively. First, the catalysts were labeled with $^{18}O$ isotope by 5 CV cycles at a scan rate of 5 mV s$^{-1}$ in 0.1 M HClO$_4$ solution containing H$_2^{18}$O. Considering the difference in activity between Si-RuO$_2$−0.1 and Com-RuO$_2$, the potential range of CV cycles was set as 1.0-1.35 V (vs. Ag/AgCl) for Si-RuO$_2$−0.1 and 1.0-1.7 V (vs. Ag/AgCl) for Com-RuO$_2$ to achieve a similar current intensity. Then, the resulting electrodes were rinsed with $^{16}O$ water several times to remove the residual H$_2^{18}$O. Finally, the electrodes were placed in 0.1 M HClO$_4$ containing H$_2^{16}$O, and CV was carried out within the above potential windows. Meanwhile, mass spectrometry was used to detect O$_2$ generated during the OER process in real time.

**Theoretical calculations**
Density functional theory (DFT) calculations were conducted via the Vienna Ab initio Simulation Package (VASP). The electronic structures of materials were described by the generalized gradient approximation (GGA) of Perdew-Burke-Ernzerhof (PBE) and the projector augmented wave (PAW). The kinetic cutoff energy of the plane wave was fixed at 450 eV. The convergence tolerance of force and energy for each atom were 0.02 eV/Å and 10$^{-5}$ eV, respectively.

A unit cell of pristine RuO$_2$ contained 48 atoms, including 16 Ru atoms and 32 O atoms. On this basis, we constructed the Ru$_{16}$Si$_2$O$_{32}$

model based on the predicted Si content in the main text. During the structural optimization process, Brillouin zone integration was performed with $3 \times 3 \times 4$ gamma k-point sampling. All atom and lattice parameters were free to vary. According to the principle of energy minimization, the most stable structure was selected to perform subsequent calculations. Detailed information on the modeled structures after optimization is provided in Supplementary Fig. 8-9 and Supplementary Table 2.

For the slab model, pristine $RuO_2$ had a four-layer Ru-O structure, namely, 64 Ru atoms and 120 O atoms. The Si-$RuO_2$ model contained four additional silicon atoms in the lattice interstices. The top two layers of Ru-O structures were relaxed, and the bottom two layers of Ru-O structures were set to be static to simulate the surface relaxation. Monkhorst-Pack k-point sampling ($2 \times 2 \times 1$ and $4 \times 4 \times 1$) was applied for geometric optimization and density of states (DOS) calculations.

To evaluate catalyst activity, models of the reaction intermediates (*OH, *O and *OOH) adsorbed onto the $RuO_2$ and Si-$RuO_2$ catalysts were also constructed, and each model was optimized to the most stable state. The free energy ($\Delta G$) of each OER step was calculated according to the following equation:

$$\Delta G = \Delta E_{ZPE} + \Delta E - T \times \Delta S \quad (3)$$

where $\Delta E_{ZPE}$ is the zero-point energy at 298.15 K; $\Delta E$ is the binding energy of the intermediates; $T$ is the experimental temperature (298.15 K); and $\Delta S$ is the entropy change. Detailed information is present in Supplementary Fig. 11 and Supplementary Table 3.

## Data availability
All relevant data generated in this study are provided in the Supplementary Information/Source Data file.

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

## Acknowledgements

This study was financially supported by the National Key R&D Program of China (2020YFB1506002, S.C.), the National Natural Science Foundation of China (Grant Nos. 22178034, 21978028, S.C.; Grant Nos. 52021004, 91834301, Z.W.) and the Chongqing Talent Program (cstc2022ycjh-bgzxm0096, S.C.).

## Author contributions

S.C. and Z.W. conceived the project. S.C. directed the main experimental works. S.C. and X.P. analyzed the experimental data. X.P. carried out the sample synthesis, characterization, electrochemical measurements. Y.L. carried out DFT calculation. L.Z. participated in some of the experimental work. Y. S. performed the electron-microscopy characterization. X.P., S.C., L.G. and Z.W. wrote the manuscript together.

## Competing interests

The authors declare no competing interests.
