## [Peer Review File · Nature Communications]

REVIEWER COMMENTS

Reviewer #1 (Remarks to the Author):

In the manuscript, the authors reported their experimental and computational results on the electrocatalytic activity and stability of Si-doped RuO₂ catalysts for oxygen evolution reaction (OER) in acid media. Specifically, the authors had synthesized the catalysts using cation-exchange resin pyrolysis approach, characterized the structure of the catalysts using XRD, TEM, and XPS, and measured the catalytic activity and stability of the catalysts for OER in acid media. It is noted that the authors also performed density functional theory (DFT) calculations to predict the structure of the Si-doped RuO₂ and OER reaction pathway on the catalysts. The authors concluded that the interstitial doped Si would enhance both the activity and stability of RuO₂ for OER in acid. The presented results only add some incremental knowledge/data to the current understanding of electrocatalysis. Some conclusions are questionable. This reviewer does not believe the current manuscript contain enough innovative, significant contents to be considered for publication in Nature Communications.

My major criticisms to the manuscript are given in below.

1. It is unclear what the significances of this presented work are. (1) Some recent work reported much better OER activity and durability of RuO₂ through doping than the presented catalysts. For example, "Non-iridium-based electrocatalyst for durable acidic oxygen evolution reaction in proton exchange membrane water electrolysis", Nature Materials, 2023. (2) The employed experimental and computational techniques are widely used in the catalyst study. (3) The authors could not solidly prove that all the Si dopants are in the assumed interstitial positions. (4) Not much new knowledge is generated from this work.
2. The structural characterization results in Figure 1 are not sufficient enough to directly confirm that the Si dopants lie in the interstitial locations of RuO₂ crystal.
3. The computational results in Fig. 1(c) are questionable. First, no detailed information of the modelled structures is given. It is unclear if the crystal structure (including volume and shape) has been fully optimized. Secondly, the nearly linear decrease of the so-called "energy cost" with number of Si atoms would lead to an incorrect conclusion that Si would be preferred to be inserted into RuO₂ in a very large amount. The experimental data presented in this manuscript do not support this prediction.
4. The authors did not provide convincing explanation to the observed enhancement in both activity and durability of the Si-doped catalysts. First, no detailed information of the modelled structures is given. It is

unclear if the normally-assumed O-saturated surface structures were employed in these computations. All the structures and adsorption energies should be given in SI. Secondly, the presented free energy change due to Si doping is within the uncertainty of the DFT method.

Reviewer #2 (Remarks to the Author):

This manuscript reports a strategy to improve the stability of RuO₂ by introducing Si element to the lattice interstices of RuO₂. The authors attribute the excellent stability to the following three aspects: 1) higher bond dissociation energy of the Si-O bond; 2) low Ru-O bond covalency; and 3) the acid resistance of Si. It is a very interesting idea. Meanwhile, the authors provide enough evidence to support this story. To be specific, in the structural characterization part, the author first verified that Si was inserted into the RuO₂ interstices by XRD, the simulated XRD and DFT calculations. Furthermore, combined TEM-EDS and XRD results, they deduced that the ideal doping level of Si in RuO₂ interstices was about 10%. To study the Ru-O bond covalency, the authors present XPS and XAFS measurements and DFT calculations, indicating that Si elements in interstitial sites play a key role in weakening the covalency of Ru-O bonds. In the electrochemical test part, the authors demonstrated that the addition of Si not only has a positive effect on OER activity, but also inhibited the dissolution of Ru ions, resulting in excellent stability. Finally, DEMS measurement directly demonstrated that the lattice oxygen oxidation pathway was markedly suppressed due to the introduction of Si. Overall consideration, I recommend its publication in Nature Communications. However, the following points should be addressed before considering the publication of the manuscript.

Major comments:

1. On page 3, lines 104-105, the EDS mappings of Si, Ru and O and XRD patterns of Si-RuO₂-0.05, Si-RuO₂-0.1 and Si-RuO₂-0.2 are similar, why the authors declared that the ideal doping level of Si in RuO₂ interstices was around 10%.
2. The authors mentioned that the SiO₂ phase is generated when the Si content exceeds 0.1. However, no diffraction peaks corresponding to SiO₂ were observed in the XRD patterns of Si-RuO₂-0.2 and Si-RuO₂-0.3. Please explain?

3. Page 4, line 119-122, the authors need to describe the reason why the binding energy of lattice oxygen constantly shifted toward a higher binding energy with increasing Si content, while that of Ru shifted toward a lower binding energy ($x \leq 0.1$) and then remained almost unchanged ($0.1 \leq x \leq 0.3$).

4. In Figure 4a, the stability follows an order of Si-RuO₂-0.1 > Si-RuO₂-0 > Com-RuO₂. However, in Figure 4c, the Ru dissolution rate of Com-RuO₂ is significantly lower than that of Si-RuO₂-0; meanwhile, this phenomenon is also observed in Figure S11. Is there a contradiction?

5. Page 7, line 197-199, the authors mentioned the following: "Considering that Si-RuO₂-0.1 has a small particle size and porous structure compared to Com-RuO₂, we deduced that the dissolution of Ru in Si-RuO₂-0.1 could be further inhibited by increasing the particle size of Si-RuO₂-0.1." I don't understand the basis of this deduction, please give your explanation.

Minor comments:

1. In Fig 1a, the font size of "RuO₂-PDF-43-1027" is too small, please modified it.

2. Should the formula of LOM proportion be $S_{34}O_2 / (S_{34}O_2 + S_{32}O_2)$?

3. Page 9, Line 254, should the amount of TEOS be 334 μ L?

4. Page 10, Line 292, regarding the solution resistance value of 30 Ω , please provide more direct evidence.

Reviewer #3 (Remarks to the Author):

This manuscript reported a Si-doping RuO₂ with extend OER stability in acid media, the durability looks become better, however, there are too conclusion are not solid, at first all, all figures in this manuscript are obscure (low resolution), I don't think this work is suitable for the publication in the nature communication, I suggest transfer to other journal after the revisions are done as followed:

1. The authors emphasized that "Si tends to interstitially insert into the RuO₂ lattice rather than replace the Ru atoms" through XRD simulations and DFT calculations, but I think persuasive evidences should be provided such as spherical aberration corrected electron microscopy, in fact, experimental data about the existence of Si in RuO₂ is too less.

2. There is also inconsistency statement about Si interstitially insert into the RuO₂ lattice, in which XRD show shift, there are no obvious different from lattice spacing, why?
3. I think even air calcined 450°C, the carbon cannot be depleted completely, which should be related to conductivity and stability. The authors can test EDS mapping by choosing carbon elements.
4. The similar work (Adv. Sci. 2023, 2207429) should be cited and discussed.
5. In line 77 of page 3, the authors stated, "...and then almost unchanged as the Si content further increased from 0.1 to 0.3." that means when the Si is beyond 0.1, it cannot be doped into RuO₂, the extra Si will form SiO₂, if like this situation, the author should characterize how many SiO₂ formed on RuO₂, this SiO₂-RuO₂ conductivity is better than commercial RuO₂? Its activity still beyond the commercial RuO₂?
6. In Fig. 2b, there is a much large offset of Si-RuO₂-x relative to SiO₂, why?
7. After the stability test, whether the morphology, metal content and metal valence of Si-RuO₂-0.1 catalyst changed. Please supplement the series of characterization after the stability test.
8. Si prevents Ov formation should be proved by EPR and so on.
9. Si content in the catalyst needs to be further determined by ICP.
10. In Figure 2b and 2c, Com-RuO₂ should add as a comparison.

General notes:

We thank all the reviewers for their thorough and valuable comments, which has helped us improve our manuscript. We provide point-to-point responses in the manuscript for all reviewers' comments. The responses are noted in blue, and the revised parts in the manuscript are highlighted in yellow. To better respond to the reviewer's comments, some figures published or in the original manuscript are cited in this reply and relabeled as Fig.R.

Response to Reviewers' Comments

Reviewer #1 (Remarks to the Author):

In the manuscript, the authors reported their experimental and computational results on the electrocatalytic activity and stability of Si-doped RuO₂ catalysts for oxygen evolution reaction (OER) in acid media. Specifically, the authors had synthesized the catalysts using cation-exchange resin pyrolysis approach, characterized the structure of the catalysts using XRD, TEM, and XPS, and measured the catalytic activity and stability of the catalysts for OER in acid media. It is noted that the authors also performed density functional theory (DFT) calculations to predict the structure of the Si-doped RuO₂ and OER reaction pathway on the catalysts. The authors concluded that the interstitial doped Si would enhance both the activity and stability of RuO₂ for OER in acid. The presented results only add some incremental knowledge/data to the current understanding of electrocatalysis. Some conclusions are questionable. This reviewer does not believe the current manuscript contain enough innovative, significant contents to be considered for publication in Nature Communications.

Response:

Thanks for your comments and valuable suggestions.

My major criticisms to the manuscript are given in below.

1. It is unclear what the significances of this presented work are.

Response:

Improving the stability of Ru-based oxide is a long-term process for the PEM water electrolyzer because of their low cost and high activity. This work aims to provide a new strategy for approaching the highly stable Ru-based oxide toward acidic water oxidation.

(1) Some recent work reported much better OER activity and durability of RuO₂ through doping than the presented catalysts. For example, "Non-iridium-based electrocatalyst for durable acidic oxygen evolution reaction in proton exchange membrane water electrolysis", Nature Materials, 2023. (4) Not much new knowledge is generated from this work.

Response:

Many thanks for recommending the paper published in *Nature Materials*. We have carefully read and analyzed it. In the reported work, the authors presented a Ni-RuO₂ catalyst with high activity and durability in acidic OER by Ni-substituted Ru site in RuO₂. Notably, there are some noticeable differences between our work and it, such as ① the doping element was metalloid (Si); ② the doping mode was interstitial-site doping. These two characteristics break the traditional thinking by metal atoms (Co, Ni, Mn, Cu, Cr, and so on) replacing the Ru atoms, and result in excellent performance. In addition, our work focuses on how to improve the stability of RuO₂

catalysts for acidic water oxidation, which encourages to researchers to explore the stability, because in real applications, OER stability may be an even more critical factor to consider (*Joule* **5**, 1–28, July 21, (2021)). Meanwhile, we propose a new concept of locking lattice oxygen to promote the stability of RuO₂ catalysts, which is essential for improving the stability of OER catalysts, especially in acidic environments. Therefore, we believe this innovative work will arouse the considerable interest of readers.

(2) *The employed experimental and computational techniques are widely used in the catalyst study.*

Response:

Advanced characterization techniques are indeed helpful for discovering new phenomena and proposing new viewpoints. However, the experimental and computational methods employed in our work were able to demonstrate our perspectives and phenomena adequately. Moreover, these well-established experimental and computational techniques are more easily understood and accepted by the reader. Therefore, we believe that conventional experimental and computational approaches can also promote the development of science and technology.

(3) *The authors could not solidly prove that all the Si dopants are in the assumed interstitial positions.*

Response:

Thanks for your comment. The reviewer is correct. Although we observed that some Si with low imaging contrast exists in the RuO₂ interstice through spherical difference electron microscopy (Fig. R1) and confirmed that Si easily enters the interstitial sites of RuO₂ through XRD and other characterization techniques, it is challenging to verify that all the Si dopants are in the RuO₂ interstice. In our manuscript, the ideal doping level of Si in RuO₂ interstices was approximately 10%, which is only inferred from the nominal ratios in the precursor mixtures and a series of characterizations.

Fig. R1. a, HAADF-STEM image of Si-RuO₂-0.1. b-c, High-resolution HAADF-STEM image obtained from the area highlighted with purple and green in Fig. 1a. d-g, Line-scanning intensity profile obtained from the area highlighted with red lines in Fig. 1b and c.

2. The structural characterization results in Figure 1 are not sufficient enough to directly confirm that the Si dopants lie in the interstitial locations of RuO₂ crystal.

Response:

Thank you for your valuable and thoughtful comments. According to your suggestion, we performed spherical aberration-corrected electron microscopy analysis. As we predicted, some Si with low imaging contrast were observed in the RuO₂ interstice in the HAADF-STEM image (Fig. R2), suggesting the Si was successfully inserted into the RuO₂ interstice.

The corresponding discussion has been updated in the revised manuscript as follows:

To visually prove that Si was inserted into the RuO₂ interstice, spherical aberration-corrected HAADF-STEM measurements were performed. As shown in Fig. 1d, the lattice fringes with interplanar spacings of 0.318 nm and 0.254 nm were assigned to the (110) and (101) planes of rutile RuO₂, respectively. Furthermore, some isolated Si atoms with low imaging contrast, which is characteristic of light elements with lower atomic numbers, were also observed in the lattice interstices of RuO₂ (Fig. 1e-f). This assertion was confirmed by atomic line profiles analysis (Fig. 1g-j).

Fig. R2. a, HAADF-STEM image of Si-RuO₂-0.1. b-c, High-resolution HAADF-STEM image obtained from the area highlighted with purple and green in Fig. 2a. d-g, Line-scanning intensity profile obtained from the area highlighted with red lines in Fig. 2b and c.

In addition, in our manuscript, we provide multidimensional evidence to demonstrate that Si is in the RuO₂ interstice.

From a theoretical standpoint:

① **Effective ion radius.** Based on Lang's Handbook of Chemistry (15th Ed.), the effective ionic radius of Si⁴⁺ is 0.26 Å while that of Ru⁴⁺ is 0.62 Å. Their lattice strain coefficients (λ_{AB}) are approximately 58% ($\lambda_{AB} = (r_A - r_B)/r_A$). According to the Hume-Rothery rule, the solubility in a binary solid solution is limited when the atomic-size mismatch is above 15% (*Scripta Materialia.*, **206**, 114226 (2022)). In other words, when λ_{AB} is greater than 15%, it is unfavorable for the formation of substituted solid solutions. Therefore, it is difficult for Si⁴⁺ to displace Ru⁴⁺ in RuO₂ (*Chemical Physics Letters.*, **398**, 235–239 (2004); *RSC Adv.*, **5**, 74790 (2015)).

② **Coordination number.** To our knowledge, the Si invariably combines with 4-coordinated oxygen in nature (*RSC Adv.*, **5**, 74790 (2015)); however, Ru is coordinated with six oxygen atoms. If Si replaces Ru, Si is coordinated with six oxygens to form SiO₆ octahedra. It is well-known that SiO₆ octahedra are unstable under ambient conditions (*Nature* **328**, 416–417 (1987)).

From an experimental standpoint:

① **XRD patterns.** In the case of substitution, the XRD peak shifts to higher 2θ value, indicating the incorporation of smaller Si⁴⁺ into the lattice sites of Ru⁴⁺. However, in our work, the shift of the XRD peaks to lower angle with increasing Si content to 10% indicates an expansion of the crystalline lattice. This phenomenon suggests that Si⁴⁺ ions were inserted into the interstices of the RuO₂ lattice, resulting in the expansion of the crystalline lattice (*ACS Energy Lett.*, **3**, 970–978 (2018); *Nat Commun* **13**, 3784 (2022)).

② **Simulated XRD patterns.** In the simulated XRD pattern, we observed some additional peaks at 10° < 2θ < 20° in the case of substitutional doping. The reason for this phenomenon is that when Si replaces the Ru site, the crystal structure of RuO₂ suffers severe distortion because the ionic radius of Si⁴⁺ is much smaller than that of Ru⁴⁺. However, for the interstitial-doped case, no additional peaks appeared except for the corresponding peak of RuO₂, suggesting that interstitial-doping of Si cannot change the crystalline structure of RuO₂, which is consistent with the experimental XRD patterns and common in interstitial doping examples (*J Mater Sci: Mater Electron.*, **29**, 9137–9141 (2018).; *ACS Appl. Energy Mater.*, **4**, 13636–13645 (2021); *Chem. Mater.*, **33**, 4135–4145 (2021)).

③ **DFT calculations.** In our work, DFT calculations were also performed to investigate the location of Si⁴⁺ in RuO₂ from an energy point of view. Based on the calculation results, we conclude that Si ions tend to insert into RuO₂ interstice rather than replace the Ru atoms in RuO₂, which also verifies that SiO₆ octahedra are unstable.

Based on these considerations, we believe that the characterization results we present provide enough evidence that Si is doped into the interstitial locations of RuO₂.

3. The computational results in Fig. 1(c) are questionable. First, no detailed information of the modelled structures is given. It is unclear if the crystal structure (including volume and shape) has been fully optimized. Secondly, the nearly linear decrease of the so-called “energy cost” with number of Si atoms would lead to an incorrect conclusion that Si would be preferred to be inserted into RuO₂ in a very large amount. The experimental data presented in this manuscript do not support this prediction.

Response:

Thank you for your helpful suggestion. After repeated verification, we can confidently answer the reviewer that our computational results are believable. However, we apologize that we did not display detailed calculation information in the original manuscript, causing you to misunderstand. We have added the detailed calculation information in the revised manuscript.

All the calculations were performed after the crystal structure had been fully optimized. During the optimization process, we first constructed several representative structural models based on different doping amounts and doping modes (Fig. R3a-q), in which we empirically excluded unstable and repetitive models. Subsequently, the energies of different structures were calculated. Finally, the most stable structure was selected to perform subsequent operations according to the principle of lowest energy (Fig. R3r). Per your suggestion, we provided detailed information on the

modeled structures after optimization (Table.R1) and supplemented them in the revised manuscript (Supplementary Figure 1 and Table 1).

Second, this so-called "energy cost" was derived from the reported literature (*Nat. Commun.*, **13**, 3784 (2022)). As determined from our calculation results, Si insertion is exothermic and spontaneous, implying that a large amount of Si can be inserted into the RuO₂ interstice. As stated by the reviewer, our experimental data presented in this manuscript do not support this prediction. This phenomenon is because the theoretical calculation only considers their thermodynamic trends (initial state and final state) under ideal conditions. However, in practice, the experimental conditions (for example, precursor, annealing temperature and time, pressure,) will also directly affect the solubility of Si in RuO₂. Even so, we have demonstrated experimentally and theoretically that a small number of Si atoms (about 10%) are more likely to insert into the RuO₂ interstice than replace the Ru site.

Fig. R3. a-q, Structural models with different Si doping amounts and doping mode in RuO₂ (Cautions: SUB1-2 represents the second model in which a Si atom replaces the Ru site, INT2-1 represents the first model in which

two Si atoms are inserted into the RuO₂ interstice); **r**, Gibbs free energy of different structural models. The most stable structure was selected for subsequent analysis.

Table.R1. Lattice parameters and unit-cell volume of the modeled structures after optimization.

Modeled structures	Lattice parameters						Unit-cell volume (Å ³)
	a	b	c	α	β	γ	
Ru ₁₆ O ₃₂	9.03983	9.03983	6.23868	90	90	90.0021	509.815750
Interstitial doping							
Ru ₁₆ SiO ₃₂	9.16148	9.16081	6.26577	89.9999	90.0001	89.9994	525.864653
Ru ₁₆ Si ₂ O ₃₂	9.11625	9.42239	6.29591	89.9989	89.9993	90.0877	540.798275
Ru ₁₆ Si ₃ O ₃₂	9.25383	9.53314	6.31518	90.0536	90.0241	89.8362	557.110333
Ru ₁₆ Si ₄ O ₃₂	9.50991	9.52546	6.32483	89.9703	90.0423	89.4439	572.915514
Substitutional doping							
Ru ₁₅ SiO ₃₂	9.02267	9.02267	6.17226	90.0000	90.0000	90.0000	502.474729
Ru ₁₄ Si ₂ O ₃₂	8.99080	8.99080	6.12759	90.0001	89.9999	90.0192	495.320556
Ru ₁₃ Si ₃ O ₃₂	8.96932	9.00270	6.03471	90.0001	89.9999	90.1339	487.290021
Ru ₁₂ Si ₄ O ₃₂	8.88449	9.00182	5.99271	90.0000	90.0000	90.3388	479.268060

4. The authors did not provide convincing explanation to the observed enhancement in both activity and durability of the Si-doped catalysts. First, no detailed information of the modelled structures is given. It is unclear if the normally-assumed O-saturated surface structures were employed in these computations. All the structures and adsorption energies should be given in SI. Secondly, the presented free energy change due to Si doping is within the uncertainty of the DFT method.

Response:

Our experimental results are fully compatible with the theoretical calculations, so we do not understand the reviewer's claim that we did not provide a convincing explanation for the observed enhancement in both the activity and durability of the Si-doped catalysts. In our manuscript, we calculated the free energy change of reaction intermediates (*OH, *O, *OOH) on the Ru site to illustrate the effect of Si on RuO₂ activity. We found that the introduction of Si slightly lowers the energy barrier of the rate-determining step (RDS, *O → *OOH), which is consistent with our experimental results that the activity of Si-RuO₂-0.1 (an overpotential of 226 mV at 10 mA cm⁻²) is slightly better than that of Si-RuO₂-0 (an overpotential of 248 mV at 10 mA cm⁻²). Furthermore, the PDOS calculation results show that the introduced Si weakens the covalency of the Ru-O bond, which is consistent with the XPS and XAFS results, thereby enhancing stability.

First, according to the reviewer's suggestion, we provide detailed information about the modeled structures with reaction intermediates (slab, *OH, *O, *OOH) on the Ru site (Fig. R4a), which is given in the revised manuscript and supporting information. In these computations, the RuO₂ slab model is used with Ru_{CUS} (coordination unsaturated) and Ru_{BR1} (coordination saturated) on the surface. (*Energy Environ. Sci.*, **10**, 2626 (2017)). This model has been widely used to calculate the activity of RuO₂ toward the acidic OER (*Adv. Energy Mater.*, 1901313 (2019); *Nat Commun.*, **11**, 5368 (2020); *ACS Catal.*, **10**, 1152–1160 (2020); *Appl. Catal. B Environ.*, **298**, 120528 (2021); *Chem* **8**, 1673–1687, June 9, (2022)).

Second, in our manuscript, although the difference between the RDS of Si-RuO₂ (1.823 eV) and RuO₂ (1.914 eV) is only 0.091 eV, the activity difference between Si-RuO₂-0.1 (226 mV) and Si-RuO₂-0 (248 mV) is also small, indicating that the theoretical result is consistent with the

experimental results. In addition, we summarize the relationship between the energy barrier of the RDS and the activity (overpotential at 10 mA cm⁻²) reported in the literature (Table R2). The $\Delta\eta_{10}/\Delta E_{RDS}$ values in our work is within the range reported in the literature; therefore, we believe that the present free energy change is reasonable in our calculation.

Fig.R4. Theoretical calculations of acidic OER activity on the established model of RuO₂ and Si-RuO₂.

Table.R2. The relationship between the energy barrier of RDS and activity.

Catalysts	Energy barrier of RDS (eV)	ΔE_{RDS} (eV)	Overpotential (mV@10 mA cm ⁻²)	$\Delta\eta_{10}$ (mV)	$\Delta\eta_{10}/\Delta E_{RDS}$	Reference
Si-RuO₂	1.823	0.091	226	22	241.7	This work
RuO₂	1.914		248			
CaCu ₃ Ru ₄ O ₁₂	1.89		171			Nat. Commun. 10 1-7 (2019)
RuO ₂	2.08	0.19	316	145	763.2	
Cr _{0.6} Ru _{0.4} O ₂	1.87		178			Nat. Commun. 10 162 (2019)
RuO ₂	2.02	0.15	297	119	793.3	
Ufd-RuO ₂ /CC	1.83		179			Adv. Energy Mater 9 1901313 (2019)
RuO ₂	2.08	0.25	254	75	300	
W _{0.2} Er _{0.1} Ru _{0.7} O _{2-δ}	1.69		168			Nat. Commun. 11 5368 (2020)
RuO ₂	2.02	0.33	240	72	218.2	
Mn-RuO ₂	2.71		158			ACS Catal. 10 1152-1160 (2020)
RuO ₂	2.89	0.18	232	74	411.1	
Mn _{0.73} Ru _{0.27} O ₂	1.71		208			Energy Environ. Sci. 15 2356-2365 (2022)
RuO ₂	2.74	1.03	300	92	89.3	
Ru ₃ MoCeO _x	1.83		164			Appl. Catal. B Environ. 298 120528 (2021)
RuO ₂	2.01	0.18	281	117	650	
Li _{0.52} RuO ₂	1.74		156			Nat. Commun. 13 3784 (2022)
RuO ₂	2.0	0.26	320	164	630.8	

Reviewer #2 (Remarks to the Author):

This manuscript reports a strategy to improve the stability of RuO₂ by introducing Si element to the lattice interstices of RuO₂. The authors attribute the excellent stability to the following three aspects: 1) higher bond dissociation energy of the Si-O bond; 2) low Ru-O bond covalency; and 3) the acid resistance of Si. It is a very interesting idea. Meanwhile, the authors provide enough evidence to support this story. To be specific, in the structural characterization part, the author first verified that Si was inserted into the RuO₂ interstices by XRD, the simulated XRD and DFT calculations. Furthermore, combined TEM-EDS and XRD results, they deduced that the ideal doping level of Si in RuO₂ interstices was about 10%. To study the Ru-O bond covalency, the authors present XPS and XAFS measurements and DFT calculations, indicating that Si elements in interstitial sites play a key role in weakening the covalency of Ru-O bonds. In the electrochemical test part, the authors demonstrated that the addition of Si not only has a positive effect on OER activity, but also inhibited the dissolution of Ru ions, resulting in excellent stability. Finally, DEMS measurement directly demonstrated that the lattice oxygen oxidation pathway was markedly suppressed due to the introduction of Si. Overall consideration, I recommend its publication in Nature Communications. However, the following points should be addressed before considering the publication of the manuscript.

Response: We thank the reviewer for the positive comments and helpful suggestions.

Major comments:

1. On page 3, lines 104-105, the EDS mappings of Si, Ru and O and XRD patterns of Si-RuO₂-0.05, Si-RuO₂-0.1 and Si-RuO₂-0.2 are similar, why the authors declared that the ideal doping level of Si in RuO₂ interstices was around 10%.

Response:

Thanks for your comment. We list several reasons below to explain that the ideal doping level of Si in RuO₂ interstices is around 10%. ① In the XRD pattern, the 2θ value of the (110) plane no longer undergoes a significant change as the introduction of Si exceeded 10%, suggesting that the lattice of RuO₂ no longer expands and implying that the excess Si was unable to insert into the RuO₂ lattice to change the lattice parameters of RuO₂. ② As seen from the EDS spectra, Si-RuO₂-0.05, Si-RuO₂-0.1, and Si-RuO₂-0.2 samples show that Si, Ru, and O were uniformly distributed throughout the entire catalyst. In the first two samples, the segregation of Si was not observed due to the insertion of Si into the RuO₂ interstice; for Si-RuO₂-0.2, no aggregation of Si and O was observed, which may be because a small amount of SiO₂ was formed and dispersed in the sample, and EDS technology could not accurately capture the accumulation of Si and O. ③ In subsequent Si 2p XPS tests, only one characteristic peak was present in the Si-RuO₂-0.05 and Si-RuO₂-0.1 samples; however, in the Si-RuO₂-0.2 samples, an additional peak consistent with the SiO₂ phase appeared, suggesting that the SiO₂ phase was present in the Si-RuO₂-0.2 samples. Based on the above results and analysis, it is reasonable to declare that the ideal doping level of Si in the RuO₂ interstice was around 10%.

2. The authors mentioned that the SiO₂ phase is generated when the Si content exceeds 0.1. However, no diffraction peaks corresponding to SiO₂ were observed in the XRD patterns of Si-RuO₂-0.2 and Si-RuO₂-0.3. Please explain?

Response:

Thanks for the comment. We believe that there are two main reasons why no diffraction peaks corresponding to SiO₂ were observed in the XRD patterns of Si-RuO₂-0.2 and Si-RuO₂-0.3: ① Under the experimental conditions, the generated SiO₂ was amorphous and could not present an obvious diffraction peak in the XRD patterns (Fig. R5); ② Due to the small amount of Si precursor added and some Si entered the lattice of RuO₂, the formed SiO₂ was present in a trace amount. Therefore, observing diffraction peaks corresponding to SiO₂ in XRD patterns is difficult.

Fig. R5. XRD pattern of SiO₂ prepared under the same conditions

3. Page 4, line 119-122, the authors need to describe the reason why the binding energy of lattice oxygen constantly shifted toward a higher binding energy with increasing Si content, while that of Ru shifted toward a lower binding energy ($x \leq 0.1$) and then remained almost unchanged ($0.1 \leq x \leq 0.3$).

Response:

Many thanks for the helpful suggestion. With increasing Si content, the amount of Si-O bonds formed in the samples continuously increased; moreover, the binding energy of the Si-O bonds was much higher than that of the Ru-O bonds, which led to the binding energy of lattice oxygen constantly shifting toward a higher binding energy (Fig. R6). In terms of Ru, when the content of Si was lower than 0.1 ($x \leq 0.1$), Si was inserted into the interstitial site of RuO₂, which caused the binding energy of Ru to shift toward a lower binding energy. However, when the content of Si was greater than 0.1 ($0.1 \leq x \leq 0.3$), extra Si existed in the form of SiO₂, which could not change the electronic structure of Ru, so the binding energy of Ru was almost unchanged.

Fig. R6. The O1s binding energy of several typical oxygen-containing species (from *Handbook of X-ray Photoelectron Spectroscopy*).

4. In Figure 4a, the stability follows an order of $\text{Si-RuO}_2\text{-0.1} > \text{Si-RuO}_2\text{-0} > \text{Com-RuO}_2$. However, in Figure 4c, the Ru dissolution rate of Com-RuO_2 is significantly lower than that of $\text{Si-RuO}_2\text{-0}$; meanwhile, this phenomenon is also observed in Figure S11. Is there a contradiction?

Response:

Thank you for your careful consideration. It is well known that the higher the crystallinity, the lower the dissolution rate of ions, that is, the higher the stability (*Joule* **5**, 1–28, July 21, (2021); *Energy Environ. Sci.*, **12**, 3548–3555 (2019); *Nat. Catal.*, **1**, 508–515 (2018). *Angew. Chem. Int. Ed.*, **57**, 2488–2491 (2018)). Compared with $\text{Si-RuO}_2\text{-0}$, Com-RuO_2 had a higher crystallinity (Fig. R7), suggesting that the Ru dissolution rate of Com-RuO_2 was lower than that of $\text{Si-RuO}_2\text{-0}$. However, the larger Com-RuO_2 particles were more prone to peel from support materials than small $\text{Si-RuO}_2\text{-0}$ particles, leading to a shorter operation time during the stability test, thus the stability follows an order of $\text{Si-RuO}_2\text{-0.1} > \text{Si-RuO}_2\text{-0} > \text{Com-RuO}_2$ in Fig. 4a.

The S-number was calculated according to the amount of produced oxygen and dissolved Ru ion ($S\text{-number} = N_{\text{O}_2} / N_{\text{dissolved Ru}}$). Under the same current density and operation time, the amount of evolved oxygen was constant, such that the S-number was only negatively related to the amount of Ru ions dissolved. Therefore, the phenomena we observed were not inconsistent with the conclusion.

Fig. R7. XRD patterns of $\text{Si-RuO}_2\text{-0}$ and Com-RuO_2 .

5. Page 7, line 197-199, the authors mentioned the following: “Considering that $\text{Si-RuO}_2\text{-0.1}$ has a small particle size and porous structure compared to Com-RuO_2 , we deduced that the dissolution of Ru in $\text{Si-RuO}_2\text{-0.1}$ could be further inhibited by increasing the particle size of $\text{Si-RuO}_2\text{-0.1}$.” I don't understand the basis of this deduction, please give your explanation.

Response:

Thanks for your kind reminder. After double-checked the deduction, we believe that this deduction has certain limitations. Therefore, we have removed this deduction in the revised manuscript.

Minor comments:

1. In Fig 1a, the font size of “RuO₂-PDF-43-1027” is too small, please modified it.

Response:

Thanks for your suggestion. We have corrected this issue in the revised manuscript.

2. Should the formula of LOM proportion be $S^{34}O_2/(S^{34}O_2+S^{32}O_2)$?

Response:

Thanks for your valuable suggestion. This information has been updated in the revised manuscript.

3. Page 9, Line 254, should the amount of TEOS be 334 μ L?

Response:

Thank you for your careful reading of our manuscript. The errors have been corrected in the revised manuscript.

4. Page10, Line 292, regarding the solution resistance value of 30 Ω , please provide more direct evidence.

Response:

Fig. R8b converted from the changed axis of Fig. 8a (from Fig. 3d in the original manuscript). In Fig. R8b to see the solution resistance value. When the vertical axis is zero ($Z'' = 0$), the intersection points between the EIS pattern of all samples and the horizontal axis are about 30 Ω , suggesting that the solution resistance value is 30 Ω .

Fig. R8. **a**, EIS plots of different catalysts from Fig. 3d in original manuscript; **b**, magnification of Fig. 3d in the original manuscript.

Reviewer #3 (Remarks to the Author):

This manuscript reported a Si-doping RuO₂ with extend OER stability in acid media, the durability looks become better, however, there are too conclusion are not solid, at first all, all figures in this manuscript are obscure (low resolution), I don't think this work is suitable for the publication in the nature communication, I suggest transfer to other journal after the revisions are done as followed:

Response:

Thanks for your comments and valuable suggestions.

I apologize for the obscure figures in this manuscript. The obscure figures may have occurred because the PDF version you saw was converted from the original Word version, and the clear figures were compressed and became blurred during this process. In fact, all figures are 600 dpi resolution in the uploaded original Word version and meet the requirements of *Nature Communications* (300 dpi or higher resolution).

1. The authors emphasized that "Si tends to interstitially insert into the RuO₂ lattice rather than replace the Ru atoms" through XRD simulations and DFT calculations, but I think persuasive evidences should be provided such as spherical aberration corrected electron microscopy, in fact, experimental data about the existence of Si in RuO₂ is too less.

Response:

Thank you for your valuable and thoughtful comments. According to your suggestion, we performed spherical aberration-corrected electron microscopy analysis. As expected, some Si with low imaging contrast was found in the RuO₂ interstice in the HAADF-STEM image (Fig. R9), suggesting the Si was successfully inserted into RuO₂ interstice.

The corresponding discussion have been updated in the revised manuscript as follow:

To visually prove that Si was inserted into the RuO₂ interstice, spherical aberration-corrected HAADF-STEM measurements were performed. As shown in Fig. 1d, the lattice fringes with interplanar spacings of 0.318 nm and 0.254 nm were assigned to the (110) and (101) planes of rutile RuO₂, respectively. Furthermore, some isolated Si atoms with low imaging contrast, which is characteristic of light elements with lower atomic numbers, were also observed in the lattice interstices of RuO₂ (Fig. 1e-f). This assertion was confirmed by atomic line profiles analysis (Fig. 1g-j).

Fig. R9. **a**, HAADF-STEM image of Si-RuO₂-0.1. **b-c**, High-resolution HAADF-STEM image obtained from the area highlighted with purple and green in Fig. 8a. **d-g**, Line-scanning intensity profile obtained from the area highlighted with red lines in Fig. 8b and c.

In our work, there are three main aspects to confirm the presence of Si in RuO₂: ① XRD patterns. The characteristic peaks of the (110) plane shifted toward lower angles with increasing Si content to 10%, indicating an expansion of the crystalline lattice. This result suggests that Si⁴⁺ ions were inserted into the RuO₂ lattice interstices. (*ACS Energy Lett.*, **3**, 970–978 (2018); *Nat Commun* **13**, 3784 (2022)). ② EDS spectra. When the Si content was less than 0.2, Si was uniformly distributed throughout the whole sample, but when the Si content increased to 0.3, SiO₂ was formed, indicating that Si was successfully introduced into RuO₂. ③ XPS. The XPS survey spectra (Fig. R10) and Si 2p spectra directly demonstrate the successful introduction of Si into RuO₂.

Fig. R10. XPS survey of all Si-RuO₂-x samples

2. There is also inconsistency statement about Si interstitially insert into the RuO₂ lattice, in which XRD show shift, there are no obvious different from lattice spacing, why?

Response:

We greatly admire your acute insight. There are two reasons for this phenomenon: ① For XRD, when Cu is used as the radiation source, the wavelength of the X-ray is 1.542 Å ($\lambda = 1.542$ Å). According to the Bragg equation ($2d \times \sin\theta = n\lambda$), we can obtain the relationship between Δd and $\Delta\theta$, which is $\frac{\Delta d}{\Delta\theta} = \frac{n\lambda}{2} \times \frac{\cos\theta}{\sin^2\theta}$. Given that the 2θ of the RuO₂ (110) plane is about 28° ($\theta = 14^\circ$, radian is 0.244, $n=1$), we obtain that the $\frac{\Delta d}{\Delta\theta}$ value is equal to 0.043 Å, indicating that the resolution is 0.043 Å at the (110) plane of RuO₂. However, the resolution of the TEM instrument (FEI Talos F200S) was 2.5 Å, suggesting that XRD had a much higher resolution than TEM. In fact, X-rays, as electromagnetic waves with short wavelengths, have less inelastic scattering with the sample; however, electrons with a resting mass have more inelastic scattering with the sample, resulting in X-rays having higher resolution for the crystal structure than electrons³. ② the Random errors are inevitable when measuring crystal plane spacing in TEM images. Combining the two factors of resolution and measurement error, we believe that it is tough to observe such a slight difference in crystal spacing in TEM images.

3. I think even air calcined 450°C, the carbon cannot be depleted completely, which should be related to conductivity and stability. The authors can test EDS mapping by choosing carbon elements.

Response:

Thank you for your careful evaluation. According to your suggestion, we performed EDS mapping of carbon elements by SEM-EDS (**Cautions:** *Considering that the carbon film was used as the sample support in the TEM-EDS test, this will seriously interfere with the measurement of trace carbon elements in the sample*). As the reviewer said, carbon is uniformly distributed throughout the sample (Fig. R11c). However, this test method is not reasonable because a large amount of carbon-containing species in the air will adsorb to the sample's interior with a porous structure, consistent with the conclusion that the C1s peak is always present in the XPS survey.

We performed a TGA test to determine whether there was residual carbon in the sample. As shown in Fig. 11g, the TGA results revealed that the mass fraction of carbon in the sample did not exceed 0.9 %. Furthermore, the C content was determined by a carbon sulfur analyzer (LECO CS230). The test result suggests that the carbon mass fraction in the Si-RuO₂-0.1 sample is 0.0024%. The two results implied that almost no carbon present in the sample.

We made considerable effort to explore the preparation conditions for entirely removing carbon in the early exploration. Finally, we determined the optimal conditions for sample preparation. Specifically, 1.5 g of CER powder with TEOS and RuCl₃ (volume of approximately 2.4 mL) was spread in a porcelain boat (10.5 cm×4.5 cm), making its thickness approximately 0.5 mm (Fig. 11h-i). Subsequently, during the annealing process, the air was also continually injected into the muffle furnace by an air pump (1.2 L/min). These strategies are conducive to the full reaction of the CER with air. In addition, it has been reported that an annealing temperature of 450 °C can completely remove carbon (*Nat. Mater.* **22**, 100–108 (2023), *Adv. Mater.* **30**, 1801351 (2018); *Nat Commun* **10**, 4875 (2019); *Adv. Energy Mater.* **11**, 2102883 (2021)).

Fig. R11. a-f, SEM image of Si-RuO₂-0.1 and the corresponding EDX elemental maps; g, TGA and DTA curve of Si-RuO₂-0.1 sample; h, Optical image of 1.5 g CER in sample tube; i, Optical image of 1.5 g CRE spread in porcelain boat.

4. The similar work (*Adv. Sci.* 2023, 2207429) should be cited and discussed.

Response:

Thanks for your kind reminder. Similar work has been cited in the revised manuscript.

After carefully reading and comparing both, there is a fundamental difference. In this similar work, the researchers claimed the importance of the Ru-Si bond in enhancing both the activity and stability of the Si-RuO_x@C catalyst; however, we pay more attention to improving the stability of RuO₂ by introducing Si element in our work and believe that the excellent stability can be attributed to the following three factors: ① higher bond dissociation energy of the Si-O bond; ② low Ru-O bond covalency; and ③ the acid resistance of Si. Therefore, our idea is completely different from this similar work.

In addition, we still have some confusion about some of the viewpoints in this article, for example:

1) The authors mistakenly cited the EXAFS results of a RuSi alloy rather than a Ru-based oxide to confirm the formation of a Ru-Si bond (located at 2.0 Å) (Fig. R12a-b). According to the previously reported works (Fig. R12c-f), the peaks at 2.0 Å should be assigned to the Ru-Cl bond, not the Ru-Si bond. In the RuSi alloy, Ru with a negative charge bonded with Si with a positive charge to form the Ru-Si bond (Fig. R12g-h). However, both Ru and Si in the Si-RuO_x@C catalyst exhibited a positive charge, as evidenced by XANES spectra (Fig. R12i). Therefore, the formation of the Ru-Si bond is impossible in the Si-RuO_x@C catalyst.

Fig. R12. **a**, FT-EXAFS fitting curve of Si-RuO_x@C, inset: schematic model (*Adv. Sci.* **2023**, 2207429); **b**, Fourier transforms of the EXAFS spectra of RuSi, RuO₂ and Ru (*Angew. Chem. Int. Ed.* **2019**, 58, 11409); **c**, The Ru k-edge k²-weighted Fourier transform spectra for Ru foil, RuCl₃, RuO₂, and Ru-N-C (from *Nat Commun.* **2019**, 10, 4849); **d**, Ru K-edge of RuCl₃@NC-M (from *Adv. Funct. Mater.* **2020**, 30, 2000531); **e**, Radial distribution of FT EXAFS signals of RuCl₃, Ru foil, Ru-SA/Pv-CoP₂, and Ru-SA/Ru cluster/Pv-CoP₂ (from *Small.* **2022**, 18, 2106870); **f**, FT-EXAFS spectra of Ru/np-MoS₂, RuO₂, RuCl₃, and Ru foil (from *Nat Commun.* **2021**, 12, 1687). **g**, Ru 3d XPS spectra of RuSi, RuO₂ and Ru (*Angew. Chem. Int. Ed.* **2019**, 58, 11409); **h**, XANES spectra at the Ru K-edge of RuSi, RuO₂ and Ru (*Angew. Chem. Int. Ed.* **2019**, 58, 11409); **i**, Si-RuO_x@C K-edge XANES spectrum (*Adv. Sci.* **2023**, 2207429).

2) The FT-EXAFS result of the Si-RuO_x@C catalyst is inconsistent with its XRD pattern. The Si-RuO_x@C catalyst showed a different FT-EXAFS curve than the RuO₂ catalyst (Fig. R13a), and the typical Ru-O-Ru bond at 3.2 Å in RuO₂ disappeared in the Si-RuO_x@C catalyst. Interestingly, the XRD pattern confirmed that the Si-RuO_x@C catalyst mainly existed in the form of RuO₂ (Fig. R13b); however, the Ru-O-Ru local structure with di-μ-oxo and μ-oxo configurations (Fig. R13c) was not present in the Si-RuO_x@C catalyst (Fig. R13a). It is thus not believable.

Fig. R13 **a**, FT-EXAFS curves of Ru foil, Si-RuO_x@C and RuO₂ (*Adv. Sci.* **2023**, 2207429); **b**, XRD patterns of Si-RuO_x@C (*Adv. Sci.* **2023**, 2207429); **c**, Ru-O-Ru local structure with di- μ -oxo and μ -oxo configurations in RuO₂.

5. In line 77 of page 3, the authors stated, “...and then almost unchanged as the Si content further increased from 0.1 to 0.3.” that means when the Si is beyond 0.1, it cannot be doped into RuO₂, the extra Si will form SiO₂, if like this situation, the author should characterize how many SiO₂ formed on RuO₂, this SiO₂-RuO₂ conductivity is better than commercial RuO₂? Its activity still beyond the commercial RuO₂?

Response:

Thank you for your thoughtful evaluation.

It is impossible to determine the precise amount of SiO₂ is generated on RuO₂ based on current characterization techniques. Moreover, it is not significant to our work to determine the amount of SiO₂ because SiO₂ as an insulator is harmful to enhancing activity.

The conductivity of SiO₂-RuO₂ is worse than that of commercial Com-RuO₂ because SiO₂ is an insulator.

There are two main reasons why the activity of Si-RuO₂-0.2 and Si-RuO₂-0.3 samples containing SiO₂ is superior to that of Com-RuO₂: ① The Si-RuO₂-0.2 and Si-RuO₂-0.3 catalysts have smaller particles and abundant pore structures, which can provide more active sites for the OER reaction; ② Although both samples contained inert and insulative SiO₂, some Si was still inserted into RuO₂ interstices and improved the intrinsic activity of RuO₂. Hence, their activity still outperforms that of Com-RuO₂.

6. In Fig. 2b, there is a much large offset of Si-RuO_{2-x} relative to SiO₂, why?

Response:

In Si-RuO_{2-x} samples, there are abundant Ru-O bonds and a few Si-O bonds, so the Ru-O bonds mainly determine the binding energy of O. However, in SiO₂, there are only Si-O bonds. Hence, the Si-O bond entirely governs the binding energy of O. Compared with the binding energy

of the Si-O bond, the binding energy of the Ru-O bond is lower, so the O binding energy of the Si-RuO_{2-x} sample is lower than that of SiO₂ (Fig. R14). Therefore, there is a much larger offset of Si-RuO_{2-x} relative to SiO₂.

Fig. R14. The O1s binding energy of several typical oxygen-containing species (from *Handbook of X-ray Photoelectron Spectroscopy*).

7. After the stability test, whether the morphology, metal content and metal valence of Si-RuO₂-0.1 catalyst changed. Please supplement the series of characterization after the stability test.

Response:

Thanks for your comments on how to improve the paper. According to your suggestion, we supplemented the series of characterizations (e.g., XRD, TEM, XPS) after the stability test, and the corresponding descriptions have been provided in the revised manuscript and Supporting Information as follows:

Furthermore, a series of characterizations, including XRD, TEM and XPS, were performed for the spent Si-RuO₂-0.1 catalysts to investigate the structural evolution of Si-RuO₂-0.1. Obviously, the crystalline structure and morphology of Si-RuO₂-0.1 still maintained its integrity after the 800h stability test (Fig. 5a-b and Supplementary Fig. 13). Meanwhile, Si, Ru and O were also uniformly distributed in Si-RuO₂-0.1, further illustrating the excellent stability of Si-RuO₂-0.1 toward acidic OER (Fig. 5c). To prove that the introduction of Si highly improved dissolution and oxidation resistance of RuO₂ toward the acidic OER, the chemical state changes for Ru and O in Si-RuO₂-0.1 before and after the 24 h stability test were further investigated and compared with those of Si-RuO₂-0 and Com-RuO₂ (**Caution:** The spent Com-RuO₂ sample was only tested for 18 h). For the Ru 3p spectra of Si-RuO₂-0.1, the Ru^{>+4}/Ru⁺⁴ value increased from 0.34 to 0.39 after the stability test, indicating the inevitable oxidation of catalysts under a high anode potential. Despite this, for the Si-RuO₂-0 and Com-RuO₂ samples, the change in the Ru^{>+4}/Ru⁺⁴ value is more significant, increasing from 0.43 to 0.58 and from 0.38 to 0.49, respectively (Fig. 5d and Supplementary Fig. 14). This result revealed that the introduced Si kept the Ru from overoxidation during the OER. Likewise, the evolution of oxygen species is also revealed by combining O 1s spectra before and after the stability test. As displayed in Fig. 5e and Supplementary Fig. 15, the remarkable increase in the O_V/O_L value for Si-RuO₂-0 and Com-RuO₂ suggests that the lattice O is involved in O₂ generation to a large extent, which will accelerate the dissolution of active Ru species. In contrast,

the O_V/O_L value was only slightly increased from 1.37 to 1.41 for Si-RuO₂-0.1, indicating that the AEM pathway dominated the OER process rather than the LOM pathway.

Fig. R15. a, XRD patterns. b, HR-TEM images, and c, corresponding mapping images for Si-RuO₂-0.1 after the 800 h OER stability test. d-e, Ru³⁺/Ru⁴⁺ and O_V/O_L ratios for Si-RuO₂-0.1, Si-RuO₂-0 and Com-RuO₂ before and after 24 h of stability. f, EPR spectra of Si-RuO₂-0.1 before and after the stability test.

8. Si prevents Ov formation should be proved by EPR and so on.

Response:

Thanks for your valuable suggestion. Per your suggestion, we performed EPR tests on the Si-RuO₂-0.1 sample before and after the stability test. As shown in Fig. R16, the EPR signal intensity at 3513 G ($g=2.001$) attributed to oxygen vacancies hardly changes, proving that Si prevents Ov formation during the OER process.

The corresponding discussion has been added to the revised manuscript as follows:

This assertion was confirmed by electron paramagnetic resonance (EPR) (Fig. 5f), in which the signal intensity of O_V at around 3513 G ($g = 2.001$) showed no obvious change.

Fig. R16. EPR spectra of Si-RuO₂-0.1 before and after the stability test.

9. Si content in the catalyst needs to be further determined by ICP.

Response:

Thank you for your concern. The Si content in representative Si-RuO₂-0.1 catalysts was determined by ICP (Table R3). The experimental results revealed that the atomic ratio of Si and Ru was close to the expected values.

Table. R3. ICP MS analysis of Si and Ru in Si-RuO₂-0.1 catalysts

Sample	Element	Sample concentration (ppm)	Si:Ru ratio
Si-RuO ₂ -0.1	Si	3302.13072	0.11: 1
	Ru	29981.1769	

10. In Figure 2b and 2c, Com-RuO₂ should add as a comparison.

Response:

Thanks for your kind reminder. The O 1s and Ru 3p spectra of Com-RuO₂ have been provided in the revised manuscript.

REVIEWER COMMENTS

Reviewer #1 (Remarks to the Author):

In the revised manuscript, the authors have made some effort to improve the quality of the presented research. However, some major issues still remain unaddressed. Consequently, this reviewer does not believe the current manuscript contain enough innovative, significant contents to be considered for publication in Nature Communications.

My major criticisms to the manuscript are given in below.

1. In the previous review report, this reviewer questioned about the significance of the presented work. Especially, recent study has already reported the same doping approach for RuO₂ catalysts (Ref: “Non-iridium-based electrocatalyst for durable acidic oxygen evolution reaction in proton exchange membrane water electrolysis”, Nature Materials, 2023). In the response letter, the authors argued that their study is distinct due to ① the doping element was metalloid (Si) and ② the doping mode was interstitial-site doping (note: this point is not sufficiently proved in the manuscript.). In my evaluation, this is a typical example of “incremental” research and thus lacks innovative contents for high impact journals.
2. In the previous review report, this reviewer pointed out that the presented computational results in Fig. 1(c) are questionable. The authors did not address this issue in the revised manuscript. The authors are suggested to consider removing these misleading results from the manuscript. As shown in Supplementary Table 1, the volume change could be as high as 12% in the computations. As real materials, few oxides can withstand this large volume change without being broken into pieces. Therefore, the presented computational results are unrealistic and could mislead the general readers.
3. In the previous report, this reviewer suggested that the authors should provide all the relaxed structures and calculated adsorption energies in SI. This will help the readers to examine the reliability of the predictions. The authors did not give all the information in this version of the manuscript. In addition, this reviewer pointed out that “the presented free energy change due to Si doping is within the uncertainty of the DFT method.”. The authors failed to address this issue in the revised manuscript.

Reviewer #2 (Remarks to the Author):

The current version is suggested to be accepted.

Reviewer #3 (Remarks to the Author):

It is suitable for the publication after authors' revision

General Notes:

We thank all the reviewers for their thorough and valuable feedback, which has helped us improve our manuscript. For all reviewers' comments, we provide point-to-point response and/or revision in original manuscript. All the reviewer comments are *italicized*, our responses are noted in blue, and the changes in manuscript are highlighted in yellow. To better respond to the reviewer's comments, some Figure published are cited in this reply and re-labeled as Fig.R.

Response to Reviewers' Comments

Reviewer #1 (Remarks to the Author):

In the revised manuscript, the authors have made some effort to improve the quality of the presented research. However, some major issues still remain unaddressed. Consequently, this reviewer does not believe the current manuscript contain enough innovative, significant contents to be considered for publication in Nature Communications.

Thanks for your comments and valuable suggestions.

My major criticisms to the manuscript are given in below.

1. In the previous review report, this reviewer questioned about the significance of the presented work. Especially, recent study has already reported the same doping approach for RuO₂ catalysts (Ref: "Non-iridium-based electrocatalyst for durable acidic oxygen evolution reaction in proton exchange membrane water electrolysis", Nature Materials, 2023). In the response letter, the authors argued that their study is distinct due to ① the doping element was metalloid (Si) and ② the doping mode was interstitial-site doping (note: this point is not sufficiently proved in the manuscript.). In my evaluation, this is a typical example of "incremental" research and thus lacks innovative contents for high impact journals.

Response:

In the last review report, we have pointed out the differences between our work and the reported work, and here we want to highlight it again in detail.

(1) Conventional dopants are usually dominated by metal atoms, such as Co, Ni, Mn, Cu, Cr, and so on. However, in our work, we chose the metalloid Si as the dopant, which is completely different from previously reported work on metal atoms as dopants.

(2) For the work published in *Nature Materials*, Ni atoms as dopant replace the Ru position in the RuO₂ lattice, and this doping mode is called substitutional doping (Fig. R1a). In our work, we reported an interstitial doping strategy (Fig. R1b), and solidly confirmed that Si is inserted into RuO₂ interstices by spherical electron microscopy (Fig. R1c-i) and so on. Therefore, in terms of doping methods, they are completely different.

To our best knowledge, the two characteristics have not been reported before. Hence, this work is enough innovative.

Fig. R1 Structural model of **a**, Ni-RuO₂ (from Fig. 4b in *Nature Materials*, 2023) and **b**, Si-RuO₂ (in our manuscript); **c**, HAADF-STEM image of Si-RuO₂-0.1. **d-e**, High-resolution HAADF-STEM image obtained from the area highlighted with purple and green in Fig. 1c. **f-i**, Line-scanning intensity profile obtained from the area highlighted with red lines in Fig. 1d and e.

2. In the previous review report, this reviewer pointed out that the presented computational results in Fig. 1(c) are questionable. The authors did not address this issue in the revised manuscript. The authors are suggested to consider removing these misleading results from the manuscript. As shown in Supplementary Table 1, the volume change could be as high as 12% in the computations. As real materials, few oxides can withstand this large volume change without being broken into pieces. Therefore, the presented computational results are unrealistic and could mislead the general readers.

Response:

Thank you for your valuable and thoughtful comments. After repeated verification, we agree the reviewer's opinion. The Fig. 1c, Supplementary Table 1 and corresponding description in original manuscript have been removed.

3. In the previous report, this reviewer suggested that the authors should provide all the relaxed

structures and calculated adsorption energies in SI. This will help the readers to examine the reliability of the predictions. The authors did not give all the information in this version of the manuscript. In addition, this reviewer pointed out that “the presented free energy change due to Si doping is within the uncertainty of the DFT method.”. The authors failed to address this issue in the revised manuscript.

Response:

According to your suggestions, all possible $\text{Ru}_{16}\text{Si}_2\text{O}_{32}$ model have been reconstructed and optimized based on the predicted Si content in the main text (the molar ratio of Si and Ru is about 10%). In detail, a Si atom first occupies A-site and serves as the reference position (Fig. R2). Next, another Si atom can be placed at one of the A, B, C and D-site. There is only one case for A-site, whereas there are two cases for B, C, and D-site (top or bottom). Then, the energies of different structures were calculated. Finally, based on the principle of lowest energy, the most stable structure (A-C-b model) was selected to perform subsequent operations (Fig. R2f and Fig. R3). The detailed information on the modeled structures after optimization (Fig. R2 and Table R1) have been provided in the revised manuscript (Supplementary Figure 8, Supplementary Figure 9 and Supplementary Table 2).

Fig. R2. The modeled structures of **a**, $\text{Ru}_{16}\text{O}_{32}$ and **b-h**, different $\text{Ru}_{16}\text{Si}_2\text{O}_{32}$ after optimization (denoted A-M-X, where A represents the first Si atom occupies the A-site; M represents the position occupied by the second Si atom, M = A, B, C or D; X represents the spatial position located by second Si atom, X = t or b, t is top, b is bottom).

Fig. R3 Gibbs free energy of different structural models.

Table R1. Lattice parameters and unit-cell volume of the modeled structures after optimization.

Modeled structures	Lattice parameters						Unit-cell volume (\AA^3)
	a	b	c	α	β	γ	
Ru ₁₆ O ₃₂	9.03983	9.03983	6.23868	90	90	90.0021	509.8158
A-A	9.31716	9.31966	6.25462	90.0054	89.9996	89.9945	543.1056
A-B-t	9.28840	9.28903	6.28375	89.9643	90.0362	90.0157	542.1636
A-B-b	9.22747	9.24128	6.32503	89.9737	90.1670	90.0108	539.3540
A-C-t	9.27393	9.27411	6.30608	90.0035	90.0035	90.0022	542.3695
A-C-b	9.28328	9.28311	6.29862	89.9967	90.0003	89.9999	542.8007
A-D-t	9.30198	9.25337	6.29996	89.9988	90.0006	89.5885	542.2534
A-D-b	9.38953	9.12285	6.31181	89.9984	90.0018	90.0746	540.6645

According to the optimized bulk structure, the partial density of states (PDOS) calculations were first carried out on O atoms and the unsaturated Ru on Ru₁₆O₃₂ and Ru₁₆Si₂O₃₂ (110) plane (Notes: Based on previous reports, the unsaturated Ru sites have been considered to be catalytic active centers, and (110) plane has been identified as the most stable surface in RuO₂). The calculation results shown that, after introducing Si into the interstitial sites, the surface Ru 4d band center (ϵ_d) upshifted from -1.759 eV to -1.626 eV, while the surface O 2p band center (ϵ_p) downshifted from -3.365 eV to -3.850 eV, suggesting that the gap between ϵ_d and ϵ_p is obviously enlarged (Fig. R3a). Further, we performed the PDOS calculation on their bulk (Fig. R4). Although the band center of Ru 4d on the bulk and the (110) plane present different trends, the gap between Ru 4d band center and O 2p band center is still enlarged due to the downshift of the O 2p band center (Fig. R4). This is still consistent with the conclusion mentioned in the original manuscript that the introduction of Si weakens the covalency of Ru-O bonds. The calculated results have been revised and the detailed information have been provided in the revised manuscript (Figure. 2f and Supplementary Figure 10).

**Fig. R4** PDOS plots of the Ru 4d and O 2p orbitals of **a**, Si-RuO₂ slab and RuO₂ slab and **b**, Si-RuO₂ bulk and RuO₂ bulk.

Finally, we recalculated the free energy change of reaction intermediates (*OH, *O, *OOH) on the unsaturated Ru site (bound to five O atoms) on (110) plane based on the fresh model. The

free energy barrier for RuO₂ is 2.028 eV, which agrees well with previous calculations (*Appl. Catal. B Environ.* **298**, 120528 (2021); *Nat. Commun.* **11**, 5368 (2020); *Nat. Commun.* **10**, 162 (2019); *Adv. Mater.* **30**, 1801351 (2018)). However, Si-RuO₂ exhibited a lower free energy barrier (1.868 eV). The calculated results (Fig. R5, R6 and Table R2) have been revised and the detailed information have been provided in the revised manuscript (Figure. 3f, Supplementary Figure 11 and Supplementary Table 3).

Fig. R5 Calculated energy barrier diagrams of Si-RuO₂ and RuO₂.

Fig. R6 Theoretical calculations of the acidic OER activity of the established models of on **a**, Si-RuO₂ and **b**, RuO₂.

Table R2 The detailed information for the free energy change of reaction intermediates (*OH, *O, *OOH) on the unsaturated Ru site.

Structure	Model	E/eV	G(T)/eV	G	Process	G-final	G (U=0 V)	ΔG
RuO ₂	Slab	-1343.076	0	-1343.076	2H ₂ O	-1371.510	0	
	*OH	-1353.752	0.320	-1353.432	H ₂ O + 1/2 H ₂	-1371.055	0.455	0.455

	*O	-1349.038	0.050	-1348.988	H ₂ O + H ₂	-1370.016	1.494	1.039
	*OOH	-1358.144	0.373	-1357.771	3/2 H ₂	-1367.988	3.522	2.028
	Slab	-1343.076	0	-1343.076	2 H ₂ + O ₂	-1366.590	4.920	1.398
	Slab	-1394.780	0	-1394.780	2H ₂ O	-1423.215	0	
	*OH	-1405.598	0.335	-1405.263	H ₂ O + 1/2 H ₂	-1422.886	0.329	0.329
Si-RuO₂	*O	-1400.770	0.054	-1400.716	H ₂ O + H ₂	-1421.744	1.470	1.141
	*OOH	-1410.033	0.374	-1409.659	3/2 H ₂	-1419.876	3.338	1.868
	Slab	-1394.780	0	-1394.780	2 H ₂ + O ₂	-1418.295	4.920	1.582

REVIEWERS' COMMENTS

Reviewer #1 (Remarks to the Author):

In the revised manuscript, the authors have provided significantly more detailed results to support the main conclusion of the study. The quality of the manuscript has been improved. Therefore, this reviewer recommends the current version of the manuscript to be considered for publication.